# Flow-induced crystallisation of polymers from aqueous solution

Gary J. Dunderdale[1], Sarah J. Davidson[1,2], Anthony J. Ryan[1] & Oleksandr O. Mykhaylyk [1✉]

Synthetic polymers are thoroughly embedded in the modern society and their consumption grows annually. Efficient routes to their production and processing have never been more important. In this respect, silk protein fibrillation is superior to conventional polymer processing, not only by achieving outstanding physical properties of materials, such as high tensile strength and toughness, but also improved process energy efficiency. Natural silk solidifies in response to flow of the liquid using conformation-dependent intermolecular interactions to desolvate (denature) protein chains. This mechanism is reproduced here by an aqueous poly(ethylene oxide) (PEO) solution, which solidifies at ambient conditions when subjected to flow. The transition requires that an energy threshold is exceeded by the flow conditions, which disrupts a protective hydration shell around polymer molecules, releasing them from a metastable state into the thermodynamically favoured crystalline state. This mechanism requires vastly lower energy inputs and demonstrates an alternative route for polymer processing.

---

[1] Department of Chemistry, The University of Sheffield, Sheffield S3 7HF, UK. [2] Croda International Plc, Snaith, Goole DN14 9AA, UK.
✉email: o.mykhaylyk@sheffield.ac.uk

In the natural world numerous strategies for efficient processing of materials have evolved[1,2], and one such solution has been recently highlighted in natural silk spinning[3,4]. Spiders and silk worms are able to extrude an aqueous polymer solution, a liquid silk dope, which solidifies to form functional structures such as webs and cocoons. Silk is widely known to have special properties such as the combination of high tensile strength, durability and biocompatibility[5], but infrequently mentioned is its ability to denature, that is convert from liquid to solid, triggered by flow. This unique property gives the animal the ability to create solid fibres from liquid silk dope, stored inside its body, in a much more energetically efficient way than fibres made from synthetic thermoplastics. In order to create crystal nuclei by shear flow, a certain amount of specific mechanical work must be performed on a polymer melt[6,7]. Silk behaves in a similar fashion, although in vitro measurements show that the specific work required to convert silk from liquid to solid by using just flow is orders of magnitude smaller than that of thermoplastics and the whole process takes place at ambient conditions[3]. In addition, there is an indication that animals can speed up the nucleation step by a careful control of the pH and ion concentration in the processing environment[8]. When extruded through a spinning duct into a fibre, the solidification is not by the commonly encountered mechanisms of heat transfer or crosslinking, specifically neither cooling nor chemical reaction, it is solely converted from one phase to another by the application of flow displacing the hydration layer surrounding silk protein molecules. To date no other material has been reported which can reproduce this mechanism under ambient conditions.

Silk protein solidification is thought to be dependent on an energetically bound shell of water molecules preventing hydrophobic regions of individual proteins from intermolecular hydrogen bonding[9]. If this hydration shell is disturbed by flow stretching the molecules during spinning, a phase transition is facilitated by the formation of intermolecular hydrogen bonds. Flow removes the water layer by changing the conformational order of proteins and this facilitates inter-protein interactions. The term aquamelt was coined to describe materials with this type of behaviour[3], but in general terms this material could be classified as a metastable aqueous polymer solution. The bound water layer plays a crucial role by keeping hydrophobic domains separated from each other in a liquid metastable state, which is then converted under the flow into a thermodynamically stable solid phase. Although this process takes place in water at ambient conditions, the resulting solid is water-insoluble with a melting

point, $T_m$, as high as 257 °C[10] corresponding to the crystallised peptide beta-sheets.

It is well-established that poly(ethylene oxide) (PEO) molecules in aqueous solutions are surrounded by a hydration layer similar to proteins[11,12]. Moreover, it has recently been demonstrated by molecular dynamic (MD) simulations that stretching of oligomer PEO chains dissolved in water initiates interchain aggregation, which ultimately leads to the phase separation of the PEO solution with the formation of highly oriented fibrillar nanostructures[13]. The aggregation was related to the change of PEO conformation making specific hydrogen-bond-induced solvation of PEO in water unfavourable which destroys the hydration layer. In this respect some observations of PEO fibrillation from aqueous solution under strong flows[14–16], previously assumed to be driven by PEO and water phase separation, could be explained by these recent MD simulations[13]. Theoretical results indicate that solidification of PEO in water solution can be triggered by stretching in analogy with silk protein dopes.

This work is inspired by the observation that silk solutions require orders of magnitude less work to induce crystallisation compared to conventional polymer melts, it tests the hypothesis that this behaviour is not unique to silk proteins but a feature of polymer solutions with specific interactions, e.g., hydrogen bonds. Using rheological properties of PEO and structural techniques based on birefringence and X-ray scattering, it is demonstrated herein that a simple synthetic polymer, PEO, with a conformation-dependent hydration layer[11–13], can be solidified and crystallised upon flow. The processing conditions required for flow-induced nucleation and crystallisation of the polymer are quantified, and related to the molecular relaxation times.

## Results

**A metastable hydration shell.** PEO is crystallisable and water soluble, due to the similarity of its oxygen–oxygen spacing to that seen in liquid water molecules[12], so also demonstrate the properties of a metastable aqueous polymer solution. The hydrophobic methylene groups of the polymer are prevented from coming into contact with each other by a sheath of bound water (Fig. 1a), consisting of about 1.6 water molecules per PEO repeat unit[17,18], which is confirmed using differential scanning calorimetry (DSC) (Fig. 2 and see "Methods" section) and Fourier-transform infrared (FTIR) spectroscopy, where there is an effect of the bound water on the frequency of PEO ether (C–O–C) stretching band peak at 1080–1100 cm$^{-1}$ (Supplementary Fig. 1 and see

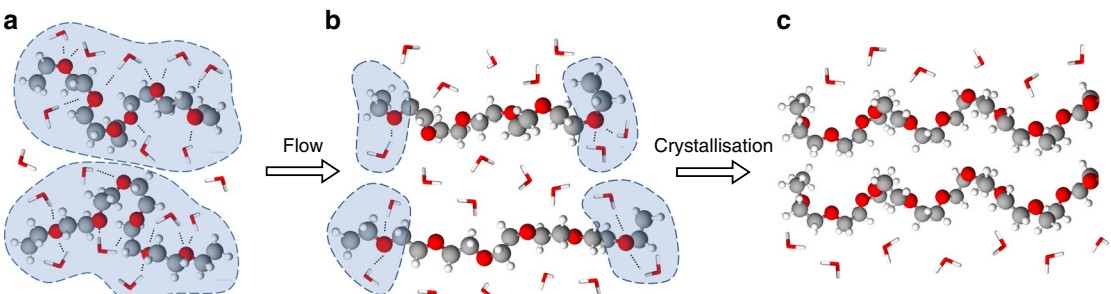

**Fig. 1 A liquid to solid phase transition of PEO in a water solution.** A schematic representation of the phase transition under flow (flow-induced crystallisation of PEO in water). In the quiescent state **a** PEO chains are coiled globular molecules (PEO segments—ball representation, water molecules—stick representation), surrounded by a protective sheath of water molecules (blue dashed lines), which prevents PEO segments from polymer intermolecular interactions. Crystallisation is prevented by this hydration sheath even when cooled far below the melting point. However, when flow is applied, **b** molecules become oriented and stretched along the flow direction leading to breakage of hydrogen bonds (dotted lines) and rupture of the hydration shell. Stretched segments of desolvated PEO, similar in orientation and conformation to that of a PEO crystal are exposed to each other. Following removal of water molecules from between the chains establishing polymer intermolecular interactions, **c** PEO chains crystallise in a helical conformation creating a solid phase.

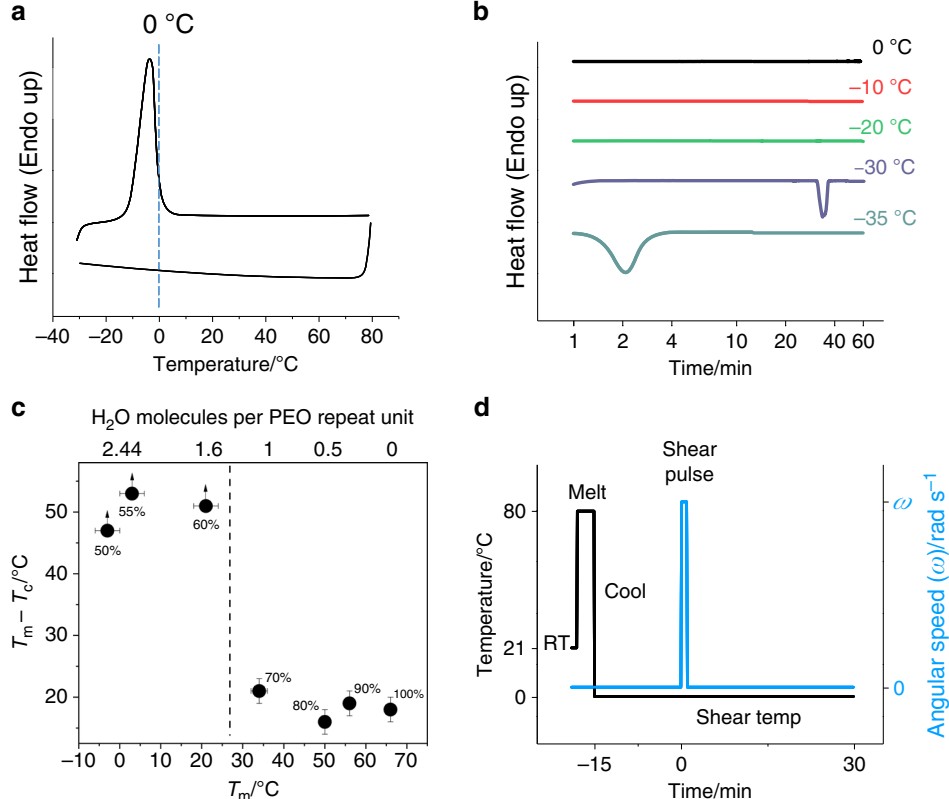

**Fig. 2 DSC of PEO solutions and shear/temperature protocol used.** Representative DSC results and shear/temperature protocol used for flow-induced nucleation of the PEO aqueous solutions. **a** Dynamic DSC measurements of 50% w/w PEO aqueous solution (heating/cooling rate 10 °C min⁻¹), the dashed line indicates a temperature of 0 °C which was chosen as the shear temperature. **b** Isothermal DSC of 50% w/w PEO aqueous solution performed at different temperatures from 0 to −35 °C. **c** Measurements of the PEO hydrated shell composition (Fig. 1a); percentage values next to the plotted symbols show w/w compositions of corresponding PEO aqueous solutions; symbols with an attached arrow indicate that crystallisation was not observed as the sample was cooled after melting, and as such these data points are an estimate of the minimum amount of hysteresis present, the horizontal error bar for each point refers to the largest error of either lower ($T_m$) or upper (H₂O molecules per PEO repeat unit) axis; dotted line represents the approximate point where the hydration shell is complete, with points to the left having more than 1.6 water molecules per PEO repeat unit, and to the right less. **d** Shear/temperature protocol used in flow-induced nucleation of 50% w/w PEO aqueous solution. Error bars, were shown, indicate the uncertainty of the measurement. (see details in "Methods" section).

"Methods" section). In molten (anhydrous) PEO the ether stretching band is observed at 1097 cm⁻¹, whereas in well-solvated dilute PEO (such as PEO in 50% w/w aqueous solution) the ether stretching band is observed at 1084 cm⁻¹ (Supplementary Fig. 1A). As water is added to PEO the peak position falls rapidly reaching the limiting solvated value at concentrations around 60% w/w (Supplementary Fig. 1B). The decrease in the C–O–C stretching frequency upon adding of water is commonly attributed to the formation of hydrogen bonds between oxygen atoms in the ether backbone of PEO and water molecules[19]. Polymer chains cannot come into close proximity without the hydration shells rupturing and de-solvation occurring. This sheath could be destabilised by a stimulus such as flow (Fig. 1b), stretching the PEO and leading to partial dehydration and aggregation of the polymer chains as predicted by MD simulations[13]. When the stretching is released the dehydrated PEO chains are likely to relax into their stable $7_2$ helical crystal structure (space group P2₁/a) because of chain flexibility and the PEO–PEO intermolecular forces[20] (Fig. 1c). This proposed mechanism is qualitatively different from the shish-kebab formation that demonstrated for polymers deposited on a free surface during the stirring of solutions of supercooled polyethylene–xylene[21,22] or PEO–ethanol[23] solutions. PEO dehydration can also be stimulated by thermal treatments. An increase of temperature decreases the solvent quality through the

reduction of hydrogen bonds; thereby PEO aqueous solutions undergo phase separation at a lower critical solution temperature of about 100 °C[24,25]. Another example is cold crystallisation of PEO[26,27], which is observed on heating preliminary cooled PEO–water mixtures, where PEO is present in a glassy state, and takes place at temperatures below solidus line of the PEO–water eutectic phase diagram[26] (about −21 °C). For these reasons the temperatures used for the shear experiments herein are above liquidus line of the PEO–water phase diagram and below 80 °C (see "Methods" section), and cannot stimulate the PEO and water phase separation without an external impact of flow.

A synthetic material crystallised by a flow (similar to a silk dope) can be created by dissolving a bimodal blend of linear PEO in water (see "Methods" section). An aqueous solution containing 50% w/w PEO exhibits a single melting transition from spherulitic crystals between −15 and +5 °C (Fig. 2a). However, the polymer solution can be held at a temperature below the spherulite melting point without crystallising instantly due to significant hysteresis. In order to create crystal nuclei and cause crystallisation under quiescent conditions, the sample had to be cooled below the solidus line[26] to −30 °C for about an hour (Fig. 2b). This hysteresis in melting/crystallisation presents an opportunity to cool the sample below $T_m$, and hold it in a meta-stable state for a significant amount of time without solidification (Fig. 1a). Then, by creating crystal nuclei through a stimulus such

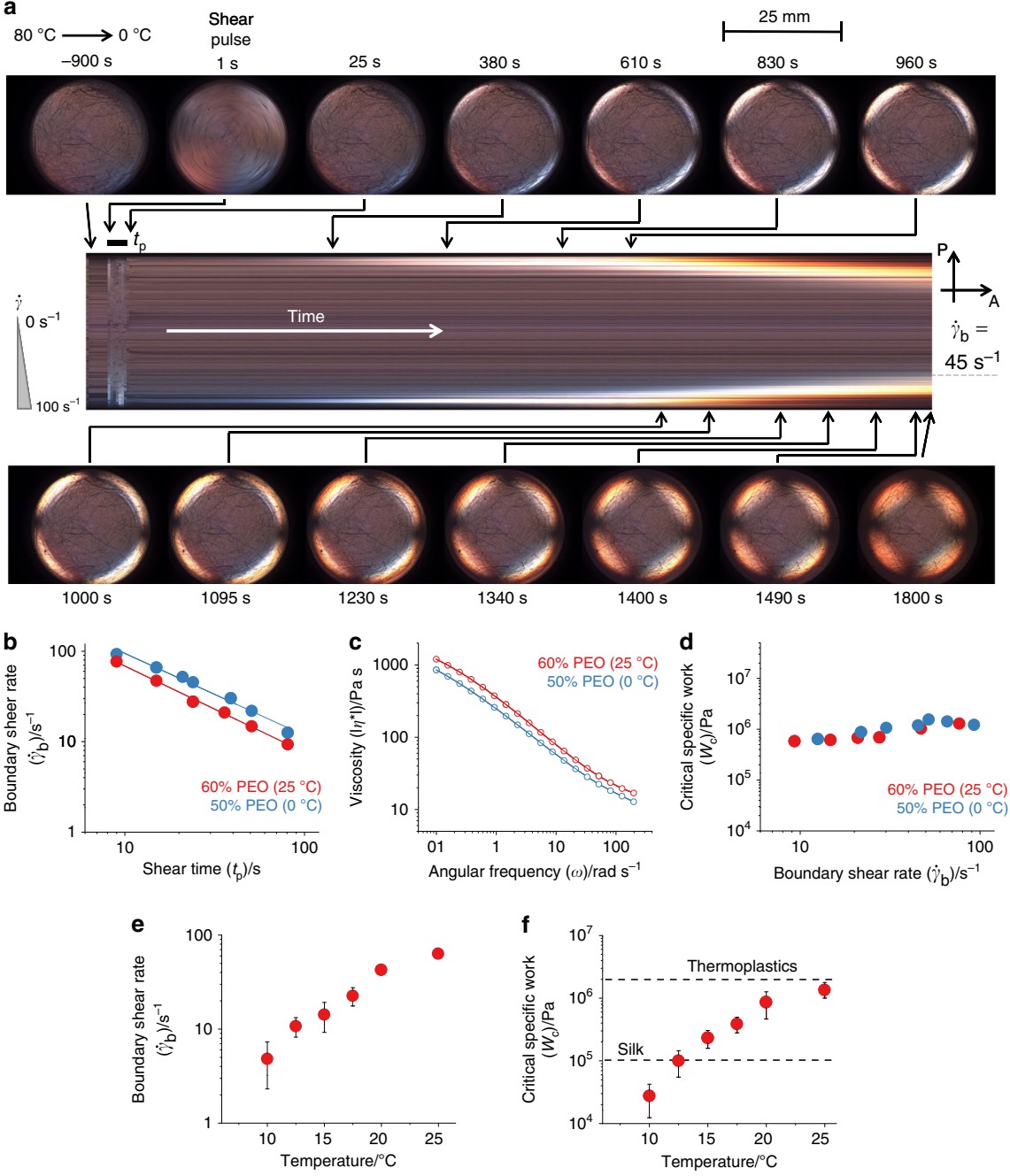

**Fig. 3 Flow conditions for nucleation of PEO in water.** Measurement of conditions required for flow-induced nucleation of PEO in water. **a** Representative polarised light images (circular images) of a sheared 50% w/w PEO aqueous solution recorded during a SIPLI experiment ($R = 12.5$ mm and SIPLI image diameter = 25 mm, $d = 0.5$ mm, $\omega = 4$ rad/s, and $t_p = 23$ s), with a time-lapse (rectangular image) composed of 45° slices through images over the course of the measurement (see "Methods" section), the vectors assigned by P and A show polariser and analyser axis, respectively. **b** A double-logarithmic plot of boundary shear rate $\dot{\gamma}_b$ measured from SIPLI experiments versus time of shearing $t_p$. **c** Magnitude of complex viscosity of the studied PEO aqueous solutions measured by a rotational rheometer using a frequency sweep mode. **d** Critical specific work $W_c$ required for flow-induced PEO nucleation versus $\dot{\gamma}_b$ detected by SIPLI. **e, f** Effect of temperature on flow-induced nucleation of 60% w/w PEO aqueous solution ($\dot{\gamma}_b$ and $W_c$, respectively) using a shear pulse of $t_p = 12$ s. Error bars, where shown, indicate uncertainty of the measurement.

as flow (Fig. 1b, c), solidify on demand. This phenomenon can be used to solidify PEO aqueous solution through flow-induced nucleation analogous to that used by silk worms and spiders. The flow can be generated in a controlled fashion by a rotational rheometer following a melt–cool cycle (Fig. 2d).

**Disruption of metastable state by flow.** A birefringence-based, shear-induced polarised light imaging (SIPLI) technique[28], previously developed to study flow-induced crystallisation of polymers[29], has been used to measure flow conditions for the nucleation of PEO in water (Fig. 3a and Supplementary Fig. 2). During a shear pulse long PEO molecules (nominal molecular weight $M_w = 2$ MDa) are stretched by the flow at shear rates greater than the inverse Rouse time ($\dot{\gamma}_R > \tau_R^{-1}$) (see "Methods" section and Supplementary Table 1) indicated by a weak Maltese Cross pattern in the polarised light image (PLI) (Fig. 3a, 1 s).

Modelling[13] and experiment[30] show that PEO chain stretching results in breakage of the bifurcated hydrogen bonds between PEO and water molecules, leading to dehydrated polymer segments (Fig. 1b). The polymers conformational entropy is also reduced upon stretching, increasing Gibbs free energy and reducing the energy barrier for the crystal nuclei formation (Fig. 1c). As the shear pulse ceases the stretched polymer chains relax and the corresponding Maltese Cross pattern fades within a fraction of second (Fig. 3a, 25 s), consequently, the sample appears as it did before the shear pulse (Fig. 3a, −900 s). However, the shear pulse has performed a certain amount of specific work on the solution allowing multiple stretched segments of the long PEO molecules to combine to form crystal nuclei[31] and this is particularly visible in the rectangular time-lapse image. Whilst these nuclei are too small and too dilute to observe optically, or by any other method, over time they grow into larger oriented polymer crystals which can be detected using polarised light. It should be noted that in the quiescent state this sample does not crystallise (at least for a day) but following a shear pulse forms oriented crystals after a few minutes. A strong Maltese Cross, indicating the formation of oriented crystals, appears on the outside edge of the image (Fig. 3a, 380 s), which grows towards the centre over time before stabilising at a certain radius (Fig. 3a, 1800 s) corresponding to the minimum (boundary) shear rate ($\dot{\gamma}_b$) required to create oriented polymer nuclei at that particular time of the shear pulse ($t_p$)[7]. Thus, in analogy to natural silks, a solid phase has been created from a metastable liquid by the application of flow (see schematic in Fig. 1).

Repeating the SIPLI experiment, converting liquid to solid, at various angular speeds and shear pulse durations, highlights the relationship between shear rate and shear time required for the formation of PEO nuclei in water. Experiments on 50% w/w PEO at 0 °C and 60% w/w at 25 °C show a boundary shear rate inversely related to the shear time (Fig. 3b) as previously demonstrated for polymer melts[7].

The critical specific work required for the PEO nucleation, $W_c$, can be calculated from $\dot{\gamma}_b$ and $t_p$ (Fig. 3b), and the magnitude of complex viscosity of the PEO solutions, $|\eta^*|$ (Fig. 3c) (see "Methods" section). Just like polyolefin melts[7,32] the work required to create crystal nuclei is independent of the shear rate or shear time used (Fig. 3d). On shallow undercooling 60% w/w and 50% w/w PEO aqueous solutions nucleate at $W_c \sim 1$ MPa (Fig. 3d), similar to the values obtained for polyolefins. However, the undercooling has a large effect on the $W_c$ for nucleation (Fig. 3e, f). Performing flow-induced nucleation at lower temperatures (greater undercooling) takes the specific work for PEO from values typical of thermoplastics ($\sim 2$ MPa)[7,32] to values similar to or even lower than those of silk ($\sim 0.1$ MPa)[3] (Fig. 3f). At such high undercooling, however, the PEO solutions are much more susceptible to thermal nucleation and have to be used in a shorter period (within hours).

In order to obtain detailed structural information, the flow-induced nucleation of 60% w/w solutions were further investigated using in-situ X-ray scattering techniques (Fig. 4). SAXS patterns show an abrupt change from isotropic weak-scattering of the initially amorphous PEO solution to highly-anisotropic strong-scattering corresponding to an oriented, semi-crystalline, lamellar morphology as shear time increased. The total scattering intensity and $P_2$ orientation function calculated from SAXS (Fig. 4 and see "Methods" section) show this change at $t_p = 60$ s, consistent with SIPLI experiments (Fig. 3), and highlight the boundary between highly oriented crystals nucleated after long shear times, and the amorphous polymer solution persistent at short shear times. Concurrent with the SAXS, WAXS patterns show a broad amorphous peak at $t_p \leq 60$ s, whereas at $t_p > 60$ s

clear Bragg peaks can be observed indicating the formation of $7_2$ helical PEO crystal structure (space group $P2_1/a$)[20]. Thus, both optical and X-ray scattering techniques produce consistent results confirming that nucleation and crystallisation of PEO from aqueous solution occurs under shear flow.

## Discussion

It was impossible to initiate crystallisation at $\dot{\gamma}$ below $\sim 10$ s$^{-1}$ in the aqueous solutions studied (Fig. 3b). This observation is consistent with the low shear-rate threshold for 2 MDa PEO stretching defined by $\dot{\gamma}_{RC}$ (Supplementary Table 1). In particular, $\dot{\gamma}_{RC}$ estimated for the PEO molecules with higher-weight-average molecular weight ($M_z = 2851$ kDa), indicative of higher molecular weight polymers present in the polymer ensemble, is similar to the experimentally detected value of the lowest shear rate resulting in PEO flow-induced nucleation (Fig. 3b). Thus, this observation confirms theoretical findings[13,30] that in order to initiate crystallisation of PEO molecules in an aqueous solution the molecules have to be stretched to remove the sheath of bound water exposing dehydrated polymer segments to each other for the crystal nucleation. There are many similarities between the flow-induced nucleation and crystallisation of polymers from aqueous solution and the flow-induced crystallisation of thermoplastics from the melt state. For polymer melts the process can be subdivided into three stages: stretching, nucleation, and alignment of the nuclei formed[33]. The stretching introduces conformational order into the polymer chains, reducing the energy barrier for nucleation, and flow delivers one stretched segment to another until they collide and form an aggregate which is larger than the critical size of a stable nucleus. Once formed, the nuclei align along the flow direction and oriented crystals grow. The same three stages are also seen in this study of aqueous polymer solutions with one crucial difference: in this case the stretching process not only induces conformational order but also removes the solvent sheath and, therefore, reduces two barriers to polymer nucleation and the subsequent crystallisation. To enable crystallisation both barriers must be affected by flow, it is not sufficient to just stretch the polymer in solution, the solvent sheath must also be removed allowing the stretched chains to aggregate.

Attempts to carry out flow-induced nucleation of the bimodal PEO blend without water, from the melt state, were complicated due to the very small undercooling range. It was impossible to initiate crystallisation of PEO by shear flow at temperatures close to, but above, the PEO peak melting point ($\geq 65$ °C). Conversely, at temperatures below the melting point (60, 62, and 63 °C) thermal nucleation dominated, and in the experimental protocol used the samples crystallised while reaching thermal equilibrium before the shear pulse. Experiments at 64 °C indicated some flow-induced nucleation (Supplementary Fig. 3), however, a combination of temperature-driven nucleation together with the shear flow produced a different morphology with a low degree of orientation (Supplementary Fig. 3 insets). These results demonstrate the importance of water in protecting PEO from thermal nucleation. The formation of an H-bonded sheath of water around each polymer chain (Fig. 1a) allows initiation of PEO crystallisation exclusively by a shear flow (Fig. 1b, c) over a much wider range of temperatures above and below the melting point of the solution.

A metastable aqueous solution of a synthetic polymer that is converted into a crystalline solid with the flow overcoming the energetic barrier to nucleation is reported. This transformation occurs under ambient conditions requiring no chemical reaction, removal of heat, or evaporation of solvent. In common with silks, the PEO solution behaves like previously reported thermoplastics,

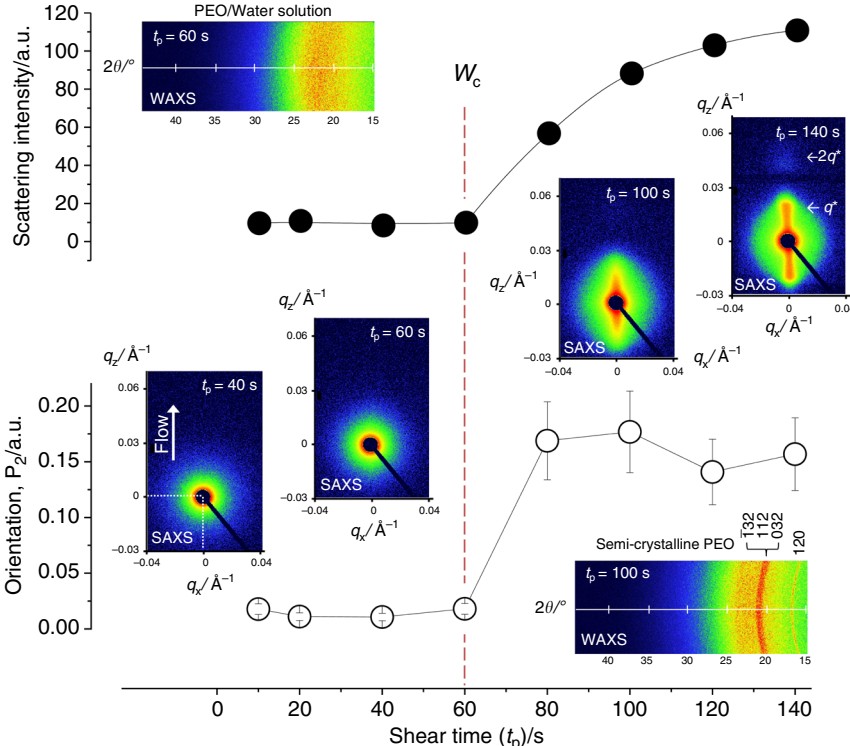

**Fig. 4 In situ rheo-SAXS/WAXS measurements of PEO crystallisation.** Characterisation of flow-induced nucleation of 60% w/w PEO aqueous solution by in situ rheo-SAXS/WAXS technique. The images are 2D scattering patterns recorded after shearing the solution at $\dot{\gamma} = 10 \text{ s}^{-1}$ for various times at 25 °C. The flow direction is shown on the SAXS pattern of $t_\text{p} = 40 \text{ s}$. The upper and lower graphs show the dependence on shearing time of the total intensity of SAXS patterns and Hermans's $P_2$ orientation function (the anisotropy calculated from 2D SAXS patterns), respectively. The red dashed line indicates the shearing time required for PEO nucleation under the chosen shear rate which was calculated from $W_\text{c}$ measured by SIPLI. Miller indexes assigned to WAXS pattern corresponding to $t_\text{p} = 100 \text{ s}$ indicate positions of diffraction peaks associated with $7_2$ helical PEO crystal structure (space group P2$_1$/a). $q^*$ and $2q^*$ shown in the SAXS pattern corresponding to $t_\text{p} = 140 \text{ s}$ indicate peak positions associated with the first and second order of lamellar morphology with the period of ~ 140 Å formed by oriented semi-crystalline PEO. Error bars indicate uncertainty associated with $P_2$ calculations.

that is, a specific amount of mechanical work needs to be performed before flow-induced nucleation can occur, and facilitate solidification. The results herein demonstrate the qualitative difference between metastable, aqueous polymer solutions, and non-polar thermoplastics, where a large window of metastability can be accessed due to the conformation-dependent solubility of water-soluble polymers[12,13,30]. The polar nature of both the polymer and solvent means that solubility is not just the non-specific effect of thermal motion (like in the regular solution model[34]) but is based on specific interactions. In this manner, one of the natural world's methods of polymer processing, flow-induced phase transitions in aqueous solutions, has been replicated using synthetic materials, leading to vastly more energy efficient polymer processing. This behaviour could be a universal phenomenon in polymer solutions that have a specific interaction that is dependent on the conformation of the polymer, allowing polymer processing with much lower energy consumption. Considering the fact that PEO aqueous solutions show a stable cold crystallisation (as a result of water molecules binding to the polymer), this phenomenon could be used as a criterion for selecting systems suitable for this method of polymer processing. A similar suggestion was made for screening biocompatible polymer systems, where a stable cold crystallisation had been proposed as an index for bound water, indicating that those polymers would be biocompatible[27].

## Methods

**Preparation and rheology of aqueous PEO bimodal blends.** Poly(ethylene oxide)s with nominal molecular weights of 2 MDa (number-average molecular weight

$M_\text{n} = 678$ kDa, weight-average molecular weight $M_\text{w} = 1799$ kDa, higher-weight-average molecular weight $M_z = 2851$ kDa and dispersity index $M_\text{w}/M_\text{n} = 2.65$, Supplementary Fig. 4) and also 20 kDa ($M_\text{n} = 19.3$ kDa, weight-average molecular weight $M_\text{w} = 21.5$ kDa, higher-weight-average molecular weight $M_z = 23.6$ kDa and dispersity index $M_\text{w}/M_\text{n} = 1.11$, Supplementary Fig. 4) were purchased from Sigma-Aldrich and used as received without further purification. Deionised water from a PureLab source with a resistivity of 18.2 MΩ was used to make aqueous PEO solutions.

In analogy with a previous research on thermoplastics[33,35] a bimodal PEO blend was used. Blending a small fraction of long polymer chains (7.5% w/w, $M_\text{w} = 2$ MDa) with a polymer matrix of short chains (92.5% w/w, $M_\text{w} = 20$ kDa) allows the polymer crystal nucleation to be triggered at a relatively small shear rate, that is above inverse Rouse time, $\dot{\gamma} > \dot{\gamma}_\text{R} = \tau_\text{R}^{-1}$, thereby stretching the long chains[7] while maintaining a reasonably low viscosity. A solvent mixing approach was used to prepare a homogenous bimodal PEO blend: the polymers were initially dissolved in water and the solvent was evaporated at a later stage.

To form aqueous PEO solutions sheets of the blended PEO (100% w/w) where cut into strips and placed inside a polypropylene disposable syringe (10 ml) followed by adding the required mass of water. The syringe tube was then sealed by attaching a plug to the needle hole before being heated to 70 °C to aid mixing. The PEO strips had visibly dissolved after a few hours. To homogenise the samples a second syringe was attached to the first and the mixture pumped back and forth between the tubes several times. The syringe tubes were then allowed to stand overnight to equilibrate the distribution of water, before being centrifuged at 4000×g for 10 min to remove any air bubbles. These aqueous solutions of bimodal PEO blends were then used in subsequent experiments.

The rheological properties of the bimodal PEO blend and its aqueous solutions were measured using a stress-controlled rheometer (Physica MCR 301, Anton Paar, Graz, Austria) in parallel-plate rotational geometry (radius of the rotating shearing disk was 12.5 mm, gap between the plates was set at 0.5 mm). A sample was loaded by applying a small amount of PEO/water mixture from a syringe to the rheometer and then melted by heating to 80 °C in the presence of a saturated water atmosphere to prevent evaporation. The heating step was used for homogenising the PEO/water mixture before the shear pulse. FTIR spectroscopy monitoring of PEO aqueous solutions at the elevated temperature indicated that there was no

effect of this treatment on the sample composition (Supplementary Fig. 1D, E). The fixture was then slowly lowered to prevent trapping air bubbles before being trimmed. A strain sweep performed at angular frequency 10 rad s$^{-1}$ confirmed linear visco-elastic behaviour up to ~10% strain, and subsequently frequency sweeps at 0.1% strain where performed at steps decreasing from high to low angular frequency (Fig. 3c and Supplementary Figs. 5 and 6). After the data collection, the magnitude of complex viscosity was fitted using a Cross model[36,37] modified as $|\eta^*(\omega)| = A_2 + \frac{A_1 - A_2}{1+(k\omega)^m}$, where $A_1$, $A_2$, $k$, and $m$ are variables (Supplementary Fig. 5, right column).

**Estimation of relaxation times of the studied PEO molecules.** Flow has two main effects on polymer behaviour in a melt or concentrated solution state: (i) an orientation of the polymer primitive path along the flow direction at shear rates $\dot{\gamma}_d > \tau_d^{-1}$, where $\tau_d$ is disengagement time associated with time required for a polymer molecule to escape a topological tube created by its surrounding chains and (ii) stretching of the polymer segments at higher shear rates $\dot{\gamma}_R > \tau_R^{-1}$, where $\tau_R$ is Rouse relaxation time. The latter is responsible for flow-induced nucleation of polymers[7,38]. The relaxation times of PEO molecules can be estimated using Likhtman–McLeish theory for linear polymers[39]: $\tau_d = 3\tau_e Z^3 (1 - \frac{3.38}{Z^{1/2}} + \frac{4.17}{Z} - \frac{1.55}{Z^{3/2}})$ and $\tau_R = \tau_e Z^2$, where $Z = M_w/M_e$ is the number of entanglements per polymer chain, and $M_e$ and $\tau_e$ are the molecular weight between entanglements and the Rouse time of an entangled polymer segment, respectively, available for PEO from literature ($M_e = 2$ kDa[40,41] and $\tau_e = 5 \times 10^{-8}$ s at 70 °C[41]). In order to estimate the PEO relaxation times at the experimental temperatures a time-temperature horizontal (frequency axis) shift coefficient obtained from Williams–Landel–Ferry (WLF) equation $\log_{10} a_T = \frac{-C_1(T - T_{ref})}{C_2 + (T - T_{ref})}$ was used. The $C_1$ and $C_2$ parameters taken from literature ($C_1 = 6.9$ and $C_2 = 88$ K at $T_{ref} = -52$ °C corresponding to PEO glass transition temperature)[42,43] were consistent with rheological data obtained for the PEO bimodal blend studied (Supplementary Fig. 6), a vertical shift coefficient was calculated from $b_T = \frac{\rho(T)(T+273.15)}{\rho(T_{ref})(T_{ref}+273.15)}$ assuming that the temperature dependence of the PEO density is expressed as $\rho(T) = \rho_0 - C_3 T$, where the polymer density at 0 °C, $\rho_0$, and $C_3$ are available from literature (1.14 g cm$^{-3}$ and 8.08 × 10$^{-4}$ g cm$^{-3}$ °C$^{-1}$, respectively)[40,43]. The WLF parameters were recalculated for $T_{ref}^{new} = 70$ °C ($C_2^{new} = C_2 + T_{ref}^{new} - T_{ref} = 210$ K and $C_1^{new} = C_1 \frac{C_2}{C_2^{new}} = 2.89$) and then applied by shifting the $\tau_e$ value to a desired temperature in order to estimate $\tau_d$ and $\tau_R$ using Likhtman–McLeish theory[39]. Finally, the effect of polymer content in the aqueous solutions was accounted for by correcting the obtained $\tau_R$ values using an empirical scaling law $\tau_{RC} = \phi^{3.5} \tau_R$ proposed for concentrated solutions[44], where $\phi$ is the polymer mass concentration (Supplementary Table 1).

**Differential scanning calorimetry of PEO materials.** By varying the concentration of aqueous PEO solutions, the melting point can be suppressed from that of pure polymer (66 °C) down to ambient temperature (~ 20 °C) for a 60% w/w solution, and down to below 0 °C for a composition of 50% w/w (Fig. 2a)[45]. The peak $T_m$ of these compositions was measured by DSC to be 66 °C, 21 °C and −2 °C, respectively using a DSC instrument (Pyris 1, Perkin-Elmer, Waltham, Massachusetts, USA) to measure the thermal response during both temperature scans and isothermal treatments. A rate of 10 °C min$^{-1}$ was used for both the heating and cooling cycles during temperature scans. A typical experiment consisted of cooling the sample to −30 °C and holding at this temperature for 5 minutes to allow crystallisation to occur, then heating to 80 °C before cooling to −30 °C once again. In order to crystallise a 50% w/w PEO aqueous solution (Fig. 2a), the hold temperature was reduced to −35 °C and the hold time increased to 10 min. Isothermal DSC of 50% w/w PEO (Fig. 2b) was performed by cooling from 25 °C to the target temperature at a rate of 10 °C min$^{-1}$, and then holding at this temperature for 60 min.

The hysteresis in DSC measurements clearly demonstrates that a hydration shell is present around PEO chains that prevents crystallisation until it is removed. If the hydration shell is absent or incomplete (less than 1.6 water molecules per monomer unit[17,18]), the difference between $T_m$ and the temperature of crystallisation, $T_c$, is consistently around 20 °C (Fig. 2c and Supplementary Fig. 7) whereas in solutions containing more than 1.6 water molecules per monomer unit (complete hydration shell), crystallisation peaks are not observed by DSC, even at temperatures 50 °C below the melting point (Fig. 2c and Supplementary Fig. 7).

**In situ polarised light imaging.** A Physica MCR 301 rheometer setup for parallel-plate rotational geometry (radius of the rotating shearing disk is 12.5 mm) is combined with an optical attachment for SIPLI to carry out the measurements. In order to visualise flow both plates of the rheometer are used as optical components: the bottom plate, made of glass, functions as a window and the top shearing plate, made of polished steel, a mirror. Thus, linear-polarised light passing through bottom glass plate is reflected from the top shearing disk, making a double pass through the sample and then passing a second polariser (analyser) crossed with the original plane of polarisation before the PLI is recorded by a CCD camera[28]. Initially a PEO solution is loaded at room temperature (~ 21 °C) before being heated to 80 °C to melt any residual polymer crystals, remove thermal history and set the shear geometry gap (usually 0.5 mm) (Fig. 2c). It is then cooled to the

desired temperature (25 °C or 0 °C for 60% w/w or 50% w/w PEO solutions, respectively) and allowed to reach thermal equilibrium over 15 min in a saturated water atmosphere (the instrument is equipped with a solvent trap) to prevent evaporation (Supplementary Fig. 8). At $t = 0$ s a rectangular shear pulse is applied to the sample by rotating the top plate at the desired angular speed (0.2 rad s$^{-1}$ ≤ $\omega$ ≤ 4.0 rad s$^{-1}$) for the desired time (9 s ≤ $t_s$ ≤ 300 s), following the cessation of flow temperature is held constant while flow-induced nucleation is further monitored using the SIPLI. Since a parallel plate geometry is used, the shear pulse generates a range of shear rates across the sample increasing radially, $\dot{\gamma} = \frac{\omega r}{d}$, where $r$ is the radial position and $d$ is the sample thickness. Thus, the sample experiences a minimum shear rate at the centre of rotation, $\dot{\gamma}_{min} = 0$ s$^{-1}$, and maximum shear rate at the edge, $\dot{\gamma}_{max} = \frac{\omega R}{d}$, where $R$ is the upper disk radius. A thermally-equilibrated PEO aqueous solution or PEO melt is thus sheared resulting in the formation of oriented PEO nuclei (not visible), which, after the cessation of shear, over time, grow into larger-oriented crystals and become visible in the polarised light as a truncated Maltese Cross around the outer part of the sample (Fig. 3a). This ring propagates towards the centre of the sample and ceases at a certain radius corresponding to the shear rate required for flow-induced nucleation[28,29]. The central non-birefringent (dark) part of the sample remains in a liquid state. A frame rate of 0.2 s$^{-1}$ has been normally used to record PLIs during shear experiments. The entire experiment can be conveniently presented as a slice of all PLIs stacked together (sliced by a plane oriented at 45° to the polariser and analyser plane), where y-axis corresponds to shear rate experienced by the sample and x-axis is time (Fig. 3a, rectangular image).

**Critical specific work for flow-induced crystallisation.** In general, the specific work performed by a flow on a sheared sample is obtained from an integral calculated over the flow time $t_p$: $W = \int_0^{t_p} \eta[\dot{\gamma}(t)]\dot{\gamma}^2(t)dt$, where is the sample viscosity represented as a function of shear rate and $\dot{\gamma}(t)$ is the shear rate as a function of time. Since a shear pulse of a rectangular shape has been used for experiments (Fig. 2c), it can be assumed that shear rate is independent of time over the pulse duration. Thus, a critical specific work value required for the PEO nucleation under shear flow conditions was calculated using a simplified equation: $W_c = \eta(\dot{\gamma}_b)\dot{\gamma}_b^2 t_p$, where $\dot{\gamma}_b$ is the shear rate corresponding to the radial position of the boundary between the crystalline and liquid parts of the sample (Fig. 3a) and $\eta(\dot{\gamma}_b)$ is the viscosity at this shear rate. Since steady-state viscosity measurements of aqueous PEO solutions at flow-induced nucleation conditions is complicated by an initiation of the polymer crystallisation, dynamic viscosity measurements have been performed instead (Fig. 3c and Supplementary Fig. 5) and the Cross model fitted to the experimental data (Fig. 3c and Supplementary Fig. 5, right column). It has been assumed for the specific work calculations that there were no transient effects due to flow and the Cox–Merz rule holds: $|\eta^*(\dot{\gamma})| = |\eta^*(\omega)|$ for $\omega = \dot{\gamma}$[37].

**In situ X-ray scattering measurements.** Small/wide-angle X-ray scattering (SAXS/WAXS) patterns were collected using a Xenocs Xeuss 2.0 laboratory beamline equipped with a high flux gallium metal jet source (Excillum, Sweden, X-ray wavelength $\lambda = 0.134$ nm), and 2D Pilatus 1 M and 100 K pixel detectors (Dectris, Switzerland). Simultaneous measurements of SAXS and WAXS were collected over a q range of 0.004 Å$^{-1}$ < $q$ < 0.3 Å$^{-1}$ and 1.19 Å$^{-1}$ < $q$ < 3.53 Å$^{-1}$ (14.5° < $2\theta$ < 44.3°), respectively, where $q = 4\pi\lambda^{-1} \sin\theta$ is the modulus of the scattering vector and $\theta$ is one-half of the scattering angle. The 2D patterns were used without the correction for background and amorphous PEO solution scattering, and radially integrated using Foxtrot software supplied with the X-ray instrument. A CSS 450 shear cell (Linkam, Tadworth, UK) fitted with steel discs with a hole (static disk) and circularly-segmented milled slots (shearing disk) and Kapton$^{TM}$ windows was used for in-situ scattering measurements. In a typical experiment, a PEO sample was loaded into a rotational parallel plate shear cell while horizontal, then the shear cell was mounted vertically onto the SAXS/WAXS laboratory beamline. The shear/temperature protocol was then performed in analogy with the SIPLI measurements (Fig. 2c) and an isothermal crystallisation allowed to proceed after a shear pulse with SAXS and WAXS collected simultaneously for 14 min with a frame rate of 1 min$^{-1}$. The formation of lamellar structure associated with PEO crystallisation was estimated from SAXS intensity calculated over q range of 0.01 Å$^{-1}$ < $q$ < 0.03 Å$^{-1}$ corresponding to the first-order diffraction peak of the lamellar structure (Fig. 4).

**Calculation of Hermans's P$_2$ orientation function.** SAXS data have been used to calculate degree of orientation of the PEO lamellar structure after flow-induced crystallisation (Fig. 4). Two dimensional SAXS patterns were azimuthally integrated over a q-range corresponding to the first-order and second-order lamella diffraction peaks, 0.01 Å$^{-1}$ < $q$ < 0.06 Å$^{-1}$, in steps of one degree. The intensity values, $I(\varphi)$, at specific azimuthal angles $\varphi$ were used to calculate Hermans's orientation function[46] defined as $P_2 = \frac{3\langle\cos^2\varphi\rangle - 1}{2}$, where $\langle\cos^2\varphi\rangle = \frac{\int_0^{\pi/2} I(\varphi)\cos^2\varphi \sin\varphi d\varphi}{\int_0^{\pi/2} I(\varphi)\sin\varphi d\varphi}$ is the average angle that the lamella normal makes with a chosen direction, which in this work is associated with the flow direction. It has also been

assumed in the expression of the average angle that there is an uniaxial orientation with symmetry around the shear direction, which enables the integration over the whole solid angle to be reduced to a quadrant. Since the experimental intensity data have a discrete distribution, analytical integration described by the formula has been replaced by numerical integration. A $P_2$ value of 1, 0, and $-0.5$ indicates orientation parallel to the flow direction, random orientation, and orientation perpendicular to the flow direction, respectively.

**FTIR spectroscopy of PEO in melt state and aqueous solutions**. In order to detect intermolecular interactions between PEO and water, infra-red spectroscopy, complementing DSC measurements (Fig. 2c and Supplementary Fig. 7), has been performed (Supplementary Fig. 1). An experimental setup has been assembled combining the environmental control of a rheometer (Physica MCR 502, Anton Paar, Graz, Austria) and an FTIR spectrometer (Nicolet iS50, Thermo Fisher Scientific), based on an approach exploited in a previously published work[47]. The instruments were coupled together by using an attenuated total reflection (ATR) accessory (Golden Gate, Specac, UK) acting on one side as the bottom plate of the rheometer parallel-plate geometry and on the other as an external sample holder of the FTIR spectrometer. The ATR accessory were incorporated in the optical path of the spectrometer via the external beam port, a set of infra-red mirrors and an external mercury cadmium telluride (MCT) detector mounted on an optical table. A PEO sample (either an aqueous solution or a bulk material) was loaded between the ATR crystal area and the top plate (a disk with a radius of 6 mm) of the rheometer equipped with a solvent trap to replicate sample environment used for the SIPLI measurements, heated to 80 °C and IR spectra were subsequently recorded (Supplementary Fig. 1A) and analysed (Supplementary Fig. 1B–E).

## Data availability

The raw data that support the findings of this study are available in figshare.shef.ac.uk with the identifier https://doi.org/10.15131/shef.data.12044556 (see ref. [48]).

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

## Acknowledgements

The authors are grateful to Masayuki Okura (Kureha Corporation, Japan) for performing the first set of measurements on flow-induced crystallisation of PEO while completing his PhD project at the University of Sheffield. O.O.M. thanks EPSRC for the capital equipment grant to purchase the laboratory-based Xenocs Xeuss 2.0/Excillum SAXS beamline used for characterizing the PEO materials (EP/M028437/1). A.J.R. and O.O.M. thank ERC for funding FLIPT project (Horizon 2020, EU project 713475). Special thanks to Sergey Donets and Jens-Uwe Sommer from The Leibniz Institute for Polymer Research (Dresden, Germany) for fruitful discussions on theoretical aspects of PEO crystallisation.

## Author contributions

A.J.R. and O.O.M. formulated the concept of this project. O.O.M. supervised the work. G.J.D. and S.J.D. who performed the measurements and analysed the experimental results. G.J.D., A.J.R., and O.O.M. jointly interpreted the results and wrote the manuscript.

## Competing interests

The authors declare no competing interests.
