## [Peer Review File · Nature Communications]

Reviewers' comments:

Reviewer #1 (Remarks to the Author):

The inspiration for this paper is the process of silk solidification under extrusion due to loss of hydrogen bonds with water and formation of intermolecular bonds assisting solidification. The objective of this paper is to show that essentially the same transition can occur for a synthetic polymer (PEO) in water under application of flow. The objective of the manuscript is attractive and novel and can be of interest for the readers of Nature Communications. The authors show that PEO/water system can be supercooled and then stimulated to crystallize by shearing. The authors applied several experimental techniques to characterize the transition and study effect of shear time on the observed phenomena. Yet, the physical origin of the observed effect remains unclear: is the observed effect analogous to silk solidification, as authors describe it or is it just cold crystallization induced by flow? It is important to get a clear answer, as the latter, i.e. cold crystallization of PEO is a well-known effect. Thus, I would recommend a major revision upon which the following points need to be addressed:

1. Authors should demonstrate that the observed effect is not cold crystallization. One way to do so is to use a different stimuli to help crystallization start, e.g. use a simple agitation (brief exposure to sonication) or addition of nucleation sites at the same supercooled conditions when authors show that shear can induce crystallization.
2. The origin of silk solidification is disruption of hydrogen bonds with water by extrusion and formation of intermolecular hydrogen bonds. PEO does not form intermolecular bonds, so it cannot act exactly as silk. The cartoon in Figure 1 shows disruption of hydrogen bonds with water, which might be accomplished, but it is less obvious why it would result in conformational change of PEO from coil to helix and what would keep the new state stable? Is there any experimental evidence that PEO forms helical conformation upon shear flow pulse application? How stable is the obtained PEO crystal? Are there any changes in scattering intensity or orientation of the crystal after a several days or longer period?
3. Figure 1 is often referred to as though it is proven by authors, but it is not. The Figure itself looks like it could be a result of a computer simulation study, but no simulation details are presented and no citation is shown in the Figure caption. It should be made clear if this Figure is just conceptual (i.e. a cartoon) or based on simulation studies. Even if it is a cartoon, the crystallized state is likely incorrect, as water plays an active role in PEO crystal formation, i.e. small water fraction helps in PEO crystallization.
4. According to Figure 2c, at low degree of supercooling the number of water molecules per repeat unit of PEO decreases and can even reach zero. This is quite surprising for 50% w/w PEO solution. How is the number of water molecules per PEO repeat unit was estimated (presumably from DSC) and what are the error-bars (and/or assumptions involved in such estimations).

Reviewer #2 (Remarks to the Author):

The authors present experiments and an argument that shear-induced crystallization in a concentrated PEO solution can be compared to the process of silk spinning from a spidroin or fibroin solution. They term this phenomenon characteristic of a "synthetic aquamelt", a contrived term that seems totally without merit or need to me (what is wrong with "an aqueous concentrated high polymer solution" which has been used for 70+ years??). The comparison with silk spinning is an interesting hypothesis and partially-supported by the data presented, but I do not feel that the present paper makes the case convincingly enough for publication in a high profile journal such as Nature Communications. The work certainly appears to

be correct, but it raises more questions than it answers, and might therefore be better suited to a sister journal such as Scientific Reports or perhaps a journal such as Soft Matter. I outline my key concerns below which I hope will help guide directions for future work:

1. One of my primary concerns is that the process of flow induced crystallization in PEO solutions is already well-known in the rheology field, yet none of this work is cited in the present paper. The authors should look at work by D.F. James & Saringer, JFM 1980 for ground-breaking classical work, as well as more recent papers by J. Eggers (Bristol), PRL 2008 and D. Bonn (Amsterdam; see for example, Deblais et al., PRL 2018). They need to clearly distinguish what makes the present work unique compared to these previous studies? For example no mention anywhere is made about flow type and the difference between kinematically-weak shear flows (as in the present work) and kinematically-strong extensional flows (which are very good at driving chain elongation).

2. Furthermore, If they are trying to make a comparison with a silk solution, then why use a bimodal blend of 20K and 2 MDa PEO chains. Is there any evidence at all that silk solutions have bimodality? If not, then why not focus on a single molecular weight of high MW PEO in a suitably viscous solvent of say oligomeric Ethylene Glycol or Glycerol (i.e. a "PEO Boger fluid" as used by Pasquali, Macosko et al. over 20 years ago, AIChE J, 1998)? Wouldn't this be much closer to a silk protein solution? If not, then please explain why not?

3. The authors also make (rather unclear conjectural) arguments on p. 8 (lines 184-190) arguing for why polydispersity in their sample can lead to crystallization to occur at shear rates less than the estimated critical value. Why not just use monodisperse samples of high MW PEO. These are readily available as GPC standards and would allow a far clearer and more complete test of the theory that a critical shear rate (or Weissenberg number) does indeed have to be exceeded...

4. In making their polymer physics arguments they make reference to the Milner/Likhtman/McLeish theory for chain-stretching in entangled solutions. This is clear and well-argued, but I wonder why they do not then compare their rheology data to the explicit prediction of the MLM theory? Why instead compare to an ad hoc empirical model (the Cross model) from the 1950s? This seems far from acceptable for a paper in NatComm?

5. Furthermore, the Cross model they use is generally used for steady shear flow data not for *linear* viscoelastic measurements of η . Why do they do this? The flow they are considering is a strong steady shear flow, and they MUST make and report steady shear flow measurements of the rheological material response for both of their concentrated polymer systems. This should be readily achievable, and would obviate the need to invoke/assume (without any supporting evidence at all) the Cox-Merz rule (line 348). Also, why use "t" as one of the model parameters, since this is shearing time in their paper??

6. Although the arguments regarding shearing strength and time make both dimensional and physical strength –and are a plausible way of computing a specific shearing energy for crystallization – there is no evidence provided that the scalings proposed make physical sense. For example, it seems to immediately suggest to me that for each data point in Figure 3 one should do a test on the same fluid at twice the imposed shear rate, for half the imposed time. The results *must* (from their arguments) superpose. Do they?

7. What about if they change the molecular weight of the long chain PEO? The Rouse time should change with M_w^2 , whereas the reptation time changes as M_w^3 (See their line 267). Which of these would best describe their experimental observations? Neither has been checked it seems to me? So why should we believe the chain stretching argument?

8. Similarly it would seem that the scaling of the variation in shear rate across the plate must be investigated directly rather than just mentioned without experimental justification: Although the

shear rate will indeed vary linearly across the plate (as per their equation on 324-326) the shear stress will NOT (because of shear thinning driven by chain alignment and shear-induced convective constraint release (as explicitly predicted by the MLM framework). Therefore the local value of the specific work W_{R} will not vary linearly with position across the disk. This scaling is not mentioned or checked by the authors...and begs a very simple test; at fixed gap separation in the parallel plate device what happens to the crystallization boundary as they increase ω ? Does it shift in linearly with rate or in a more complex fashion? This would be very straightforward to check using images like Fig 3A at different speeds

For a paper in a high profile journal like NatComm I would expect some/all of these scalings to be checked...

9. Finally, regarding the analogy with silk spinning, it has been argued by Vollrath and Knight that an active ion-exchange is involved within the spinneret itself to drive extension-induced crystallization. No mention of this is given, and it naturally raises the question what would happen in the present system if the ionic strength (and PEO confirmation) was varied (e.g. using salt solutions or brine rather than DI water?)

Reviewer #3 (Remarks to the Author):

The authors present an interesting observation on the processing of PEO from an aqueous solution where the hydration shell of water surround the PEO is disrupted by the stretching of the PEO chains during flow, a breakage of the hydrogen bonds between the water and the PEO, that, in turn, lead to dehydrated segments that can form the nuclei of subsequent crystals. This is schematically shown in Figure 1. Initially, I had thought that this was a simulation but it is only an artist's rendering. Given the inability to probe this behavior and the subsequent initial aggregation to ultimately form nuclei, it raises the question as to why simulations were not done? Given the amount of speculation that is necessary to draw the conclusions (granted from several observations that infer this behavior), I would like to have seen some simulation support for the arguments. While the threshold shear rate to induced crystallization supports theoretical arguments, these arguments are based only on the stretching of the polymer chains and are devoid of the fundamental and perhaps the most important part of this work, namely the dehydration of the PEO chains. It is that, after all, that brings in the analogy to spiders their ability to retain the silk in a dissolved state, and the air ability to spin the fibers on demand.

While I like this work overall as demonstration crystallization behavior of a hydrogen bonding polymer under shear, I am far, far less convinced of, what I consider to be hype, that this will lead to a new paradigm in polymer processing. Perhaps for a few selected, water soluble polymers but not form the preponderance of polymers that are currently processed and used on a large scale. Would the authors expect similar behavior similar behavior for conjugated polymer that exhibit ordering by intermolecular p-bonding to form H- or J-aggregates where the presence of solvents precludes interactions until a certain concentration range is reached? Here we have favorable interactions with the solvent and aggregation can occur only when the chains are in the proper conformation to interact. It's really not the same behavior but I have been trying to think of a situation where non-hydrogen bonding polymer would fall into the realm of this work. I was trying to come up with other example where such behavior could be observed. Do the authors have any suggestions to support the statement that this behavior could be a universal phenomenon, rather than simply making that statement?

So, while I like the work, I would like to see some simulations to support the arguments made and, unless the authors present other firm examples, the hype in the manuscript should be substantially reduced.

Reviewer #1 (Remarks to the Author):

The inspiration for this paper is the process of silk solidification under extrusion due to loss of hydrogen bonds with water and formation of intermolecular bonds assisting solidification. The objective of this paper is to show that essentially the same transition can occur for a synthetic polymer (PEO) in water under application of flow. The objective of the manuscript is attractive and novel and can be of interest for the readers of Nature Communications. The authors show that PEO/water system can be supercooled and then stimulated to crystallize by shearing. The authors applied several experimental techniques to characterize the transition and study effect of shear time on the observed phenomena. Yet, the physical origin of the observed effect remains unclear: is the observed effect analogous to silk solidification, as authors describe it or is it just cold crystallization induced by flow? It is important to get a clear answer, as the latter, i.e. cold crystallization of PEO is a well-known effect. Thus, I would recommend a major revision upon which the following points need to be addressed:

Reply: In our work crystallisation (solidification) of PEO is triggered by flow and in this respect the observed effect is analogous to silk solidification. We agree with the referee that cold crystallization of PEO is a known effect. However, crystallisation (and/or cold crystallisation) of PEO induced by flow is not a known effect. Recent theoretical work using MD simulations demonstrate that stretching of PEO can lead to partial dehydration and aggregation of the polymer chains and provide mechanism for phase-separation of PEO and water under an impact of the external force, this work [Donets, S. & Sommer, J.-U. Molecular Dynamics Simulations of Strain-Induced Phase Transition of Poly(ethylene oxide) in Water. *The Journal of Physical Chemistry B* **122**, 392-397] is referenced in our manuscript a few times. The simulations were performed only for a small number of PEO repeat units and PEO solidification was observed but no stable crystalline form was reached due to computational time constraints. Based on these theoretical findings, we perform systematic rheo-optical and rheo-SAXS/WAXS measurements to test the theoretical findings and demonstrate experimentally that indeed stretching of PEO molecules in water solution (by applying shear flow rate above reciprocal Rouse time of PEO molecules, in our case PEO molecules with molecular weight of about 2MDa) can induce crystallisation of PEO.

We have revised our manuscript to make this point clearer by adding an additional text including the introduction section (Page 2): "It is well-established that poly(ethylene oxide) (PEO) molecules in aqueous solutions are surrounded by a hydration layer similar to proteins.^{9,10} Moreover, it has recently been demonstrated by molecular dynamic (MD) simulations that stretching of oligomer PEO chains dissolved in water initiates the interchain aggregation, which ultimately leads to the phase separation of the PEO solution with the formation of highly oriented fibrillar nanostructures.¹¹ The aggregation was related to the change of PEO conformation making specific hydrogen-bond-induced solvation of PEO in water unfavourable which destroys the hydration layer. In this respect some previous observations of PEO fibrillation from aqueous solution under strong flows,¹²⁻¹⁴ assumed to be driven by PEO and water phase separation, could be explained by these recent MD simulations¹¹. Theoretical results indicate that solidification of PEO in water solution can be triggered by stretching in analogy with silk protein dopes. Using rheological properties of PEO and structural techniques based on birefringence and X-ray scattering, it is demonstrated herein that a simple synthetic polymer, PEO, with a conformation-dependent hydration layer,⁹⁻¹¹ can be solidified and crystallised upon flow and processing conditions required for the flow-induced nucleation and crystallisation of the polymer are quantified and related to the molecular relaxation times."

Comment: 1. Authors should demonstrate that the observed effect is not cold crystallization. One way to do so is to use a different stimuli to help crystallization start, e.g. use a simple agitation (brief

exposure to sonication) or addition of nucleation sites at the same supercooled conditions when authors show that shear can induce crystallization.

Reply: Cold crystallization is an exothermic crystallisation process which is observed on heating a preliminary cooled polymer in a glassy state. Upon heating, above the glass transition, the molecules become mobile and crystals form at a relatively low temperature. In our case the polymers are cooled down only to temperatures above liquidus line, where they are still in solution, and a controlled shear flow is used as a stimulus which causes the liquid to solid transition. This is the main novelty of our work. The mechanism is described by the recent theoretical work (Donets, S. & Sommer, J.-U.) based on the fact that the sheath of water surrounding PEO can be destabilised by stretching PEO resulting in solidification. In our experiments, after the shear pulse, the flow-induced nuclei formed by 2MDa PEO perform as nucleating agents for crystallisation of the remaining PEO. That is crystallisation of a supercooled fluid on nuclei which were created by the shear pulse. The point here is that the shear pulse induced long-lived nuclei formed by high-molecular weight 2MDa PEO and the remaining polymer, mainly 20k PEO, crystallises on them. There is no need to introduce additional nucleation sites as the studied system will eventually crystallise on its own after a few days and this has been mentioned in our original manuscript on page 5 line 122: "It should be noted that in the quiescent state this sample does not crystallise (at least for a day) but following a shear pulse forms oriented crystals after a few minutes."

Following the referee's comment, we have discussed the cold crystallisation of PEO with respect to our work in the revised manuscript (Page 3) and added four more references (refs 22-25): "PEO dehydration can also be stimulated by thermal treatments. An increase of temperature decreases the solvent quality through the reduction of hydrogen bonds; thereby PEO aqueous solutions undergo phase separation at a lower critical solution temperature of about 100 °C.^{22,23} Another example is cold crystallisation of PEO^{24,25} which is observed on heating preliminary cooled PEO-water mixtures where PEO is present in a glassy state and takes place at temperatures below solidus line of the PEO-water eutectic phase diagram²⁴ (about -21 °C). For these reasons the temperatures used for the shear experiments herein are above liquidus line of the PEO-water phase diagram and below 80 °C (**Methods**), and cannot stimulate the PEO and water phase separation without an external impact of flow."

Comment: 2. The origin of silk solidification is disruption of hydrogen bonds with water by extrusion and formation of intermolecular hydrogen bonds. PEO does not form intermolecular bonds, so it cannot act exactly as silk. The cartoon in Figure 1 show disruption of hydrogen bonds with water, which might be accomplished, but it is less obvious why it would result in conformational change of PEO from coil to helix and what would keep the new state stable? Is there any experimental evidence that PEO forms helical conformation upon shear flow pulse application? How stable is the obtained PEO crystal? Are there any changes in scattering intensity or orientation of the crystal after a several days or longer period?

Reply: We agree with the referee that "The origin of silk solidification is disruption of hydrogen bonds with water by extrusion" and this is the initial step of phase separation that defines an aquamelt. We also agree with the referee that "PEO does not form intermolecular bonds" but the initial step in both cases (silk and PEO) leads to solidification via either intermolecular hydrogen bond formation or van der Waals forces, respectively. The cartoon in Figure 1 illustrates conclusions made by recent MD simulations (Donets, S. & Sommer, J.-U.) demonstrating that the sheath of water surrounding PEO could be destabilised by stretching the PEO leading to partial dehydration and aggregation of the polymer chains. When the stretching is released the dehydrated PEO chains are likely to relax into its stable 7_2 helical crystal structure (space group $P2_1/a$) because of the chain flexibility and the PEO-PEO intermolecular forces. There is indeed no experimental evidence that the

initial PEO structure has helical conformation upon shear flow pulse application. In fact, the volume of formed nuclei under shear is tiny and its structure is undetectable. However, our measurements show that the crystals obtained are normal 7_2 helical PEO crystals formed during crystallisation under quiescent conditions on the stable nuclei created by flow. The intensity of scattering increases with time (before saturating) as the degree of crystallinity increases during annealing at temperature between crystallisation temperature and melting point. SIPLI results (Figure 3A) and X-ray scattering patterns collected at intermediate stage of annealing (Figure 4) clearly demonstrate this.

We have added further clarification in the text with regards to the MD simulations and helical structure of PEO with appropriate references (Page 3): “This sheath could be destabilised by a stimulus such as flow (Fig. 1B), stretching the PEO and leading to partial dehydration and aggregation of the polymer chains as predicted by MD simulations¹¹. When the stretching is released the dehydrated PEO chains are likely to relax into their stable 7_2 helical crystal structure (space group $P2_1/a$) because of the chain flexibility and the PEO-PEO intermolecular forces¹⁹ (Fig. 1C).”

Comment: 3. Figure 1 is often referred to as though it is proven by authors, but it is not. The Figure itself looks like it could be a result of a computer simulation study, but no simulation details are presented and no citation is shown in the Figure caption. It should be made clear if this Figure is just conceptual (i.e. a cartoon) or based on simulation studies. Even if it is a cartoon, the crystallized state is likely incorrect, as water plays an active role in PEO crystal formation, i.e. small water fraction helps in PEO crystallization.

Reply: Figure 1 is a cartoon and it is indicated at the very beginning of the figure caption “A schematic representation...”. This figure illustrates conclusions made by recent molecular dynamic simulations [Donets, S. & Sommer, J.-U. Molecular Dynamics Simulations of Strain-Induced Phase Transition of Poly(ethylene oxide) in Water. *The Journal of Physical Chemistry B* **122**, 392-397, doi:10.1021/acs.jpcc.7b10793 (2018)] demonstrating that the sheath of water surrounding PEO can be destabilised by stretching the PEO leading to partial dehydration and aggregation of the polymer chains. We have referenced this work in our original manuscript (ref 10) and cited four times in the text. Following the referee’s comment, we have extended discussion of this theoretical work in the main text and linked this discussion to Figure 1 (Page 3) to make it clear that it is based on the simulation study. The water may play a role in PEO crystal formation but it would be difficult if possible at all to localise water by the techniques we used. However, the final result is dehydrated PEO in a crystalline state. We have clarified this in the revised manuscript: “This sheath could be destabilised by a stimulus such as flow (Fig. 1B), stretching the PEO and leading to partial dehydration and aggregation of the polymer chains as predicted by MD simulations¹¹. When the stretching is released the dehydrated PEO chains are likely to relax into their stable 7_2 helical crystal structure (space group $P2_1/a$) because of the chain flexibility and the PEO-PEO intermolecular forces¹⁹ (Fig. 1C).”

Comment: 4. According to Figure 2c, at low degree of supercooling the number of water molecules per repeat unit of PEO decreases and can even reach zero. This is quite surprising for 50% w/w PEO solution. How is the number of water molecules per PEO repeat unit was estimated (presumably from DSC) and what are the error-bars (and/or assumptions involved in such estimations).

Reply: Thank you for this comment. We agree with the referee that the number of water molecules per repeat unit of PEO cannot reach zero for 50% w/w PEO aqueous solution. And this is not what we have observed in our experiment. There is a mistake in the figure caption. We have realised that the main title of the Figure 2 caption was misleading as it referenced to “50% w/w PEO aqueous

solution". This reference was correct for Figures 2A, 2B and 2D but not for Figure 2C. Figure 2C represents data for all studied compositions. This information is described in "Differential Scanning Calorimetry of PEO materials" section on page 11-12 (lines 299-305) of the original manuscript: "The hysteresis in DSC measurements clearly demonstrates that a hydration shell is present around PEO chains that prevents crystallisation until it is removed. If the hydration shell is absent or incomplete (less than 1.6 water molecules per monomer unit^{13,14}), the difference between T_m and the temperature of crystallisation, T_c , is consistently around 20 °C (Fig. 2C and Supplementary Fig. 5)." 1.6 water molecules per repeat unit of PEO corresponds to 60% w/w PEO aqueous solution (see Figure 2C in the revised version). Composition of the measured PEO-water solutions corresponding to the data points have been shown in the revised Figure 2C and appropriate amendments to the figure caption have been made in the revised manuscript. The number of water molecules per PEO repeat unit was **not** estimated from the DSC measurements. This number was calculated from the masses of water and polymer mixed together, and as such the error is small. A larger error is the measurement of T_m and T_c , and error bars have been added to reflect this, and a brief explanation added to the caption.

Reviewer #2 (Remarks to the Author):

Comment: The authors present experiments and an argument that shear-induced crystallization in a concentrated PEO solution can be compared to the process of silk spinning from a spidroin or fibroin solution. They term this phenomenon characteristic of a "synthetic aquamelt", a contrived term that seems totally without merit or need to me (what is wrong with "an aqueous concentrated high polymer solution" which has been used for 70+ years??).

Reply: We agree with the reviewer that the studied PEO water solutions can be classified as a concentrated aqueous polymer solution. There is nothing wrong with this term. In general, the aqueous polymer solutions reported in the previous 70+ years do not change from liquid to solid upon shear. We used term "aquamelt" to highlight the difference of PEO aqueous solutions in a comparison with most of other polymers because of the nature of PEO solubility in water. Protecting water layer formed around PEO molecules keeps the molecules in a solubilised metastable state which can be disrupted by flow. Based on studies of silk solidification (denaturation) in water from solubilised metastable state under flow a term "aquamelt" was coined to materials with this property. This particular property makes PEO water solution different from just an aqueous concentrated polymer solution. For example, we have tested this crucial difference by collecting data on aqueous polymer solutions such as PVA, PnVP which show no change upon shear flow. We use the term aquamelt to differentiate between a solution which solidifies upon shear, and others which do not.

Comment: The comparison with silk spinning is an interesting hypothesis and partially-supported by the data presented, but I do not feel that the present paper makes the case convincingly enough for publication in a high profile journal such as Nature Communications. The work certainly appears to be correct, but it raises more questions than it answers, and might therefore be better suited to a sister journal such as Scientific Reports or perhaps a journal such as Soft Matter. I outline my key concerns below which I hope will help guide directions for future work:

Reply: This is the first report of a class of material with very interesting properties. The fact that we are having discussions about the points highlighted in this review indicates that the results are significant and warrant publication in a journal such as Nature Communications. We have replied to

the key concerns of the referee most of which have already been addressed in our original manuscript.

Comment: 1. One of my primary concerns is that the process of flow induced crystallization in PEO solutions is already well-known in the rheology field, yet none of this work is cited in the present paper. The authors should look at work by D.F.James & Saringer, JFM 1980 for ground-breaking classical work, as well as more recent papers by J. Eggers (Bristol), PRL 2008 and D. Bonn (Amsterdam; see for example, Deblais et al., PRL 2018). They need to clearly distinguish what makes the present work unique compared to these previous studies? For example no mention anywhere is made about flow type and the difference between kinematically-weak shear flows (as in the present work) and kinematically-strong extensional flows (which are vary good at driving chain elongation).

Reply: We strongly refute the claim that flow-induced crystallization in PEO is well known, and that it has been reported before - it has not. The references the reviewer has given are not about flow-induced crystallisation! A keyword search in the papers for “crystal”, “crystallization”, or “flow induced” gives only 1 hit for the paper by D.F.James & Saringer, JFM 1980 and 0 hits in 2 of the other papers. More specifically, the first work (D. F. James & J. H. Saringer, Extensional flow of dilute polymer solutions, Journal of Fluid Mechanics, 1980, V97, 665), classified by the reviewer as a ground-breaking classical work, in fact, reports about a side observation of the formation of strands (a fibre-like morphology) where the authors just speculated that this could be a result of flow-induced crystallisation. However, there is no experimental prove of crystallisation, upstream and downstream morphology was very similar and the authors were unclear about a possible mechanism of the strand formation. The other two works cited in the reviewer report (R. Sattler, C. Wagner and J. Eggers, Blistering Pattern and Formation of Nanofibers in Capillary Thinning of Polymer Solutions, Physical Review Letters, 2008, 100, 164502 and A. Deblais, K. P. Velikov and D. Bonn, Pearling Instabilities of a Viscoelastic Thread, Physical Review Letters, 2018, 120, 194501) observe the formation of a fibre (thread) but do not even mention a single word related to crystal*, leave alone flow-induced crystallisation. Perhaps, the reviewer has an impression that the cited works report about flow-induced crystallisation but this has to be proven experimentally and this is what we have done in our work for the first time by using structural and rheology techniques commonly accepted in the field of flow-induced nucleation and crystallisation of polymers. Moreover, we do not just report on the flow-induced nucleation and crystallisation of PEO water solutions. Based on recent MD simulations and rheological properties of the material we provide a mechanism for the PEO nucleation and crystallisation. Based on the results of our work the observations presented in the papers proposed by the reviewer can be interpreted as flow-induced crystallisation but not the other way round. Since those observations could be explained by our results and support our findings, we have referenced these papers in the extended introduction of our revised manuscript (refs 12-14):

“It is well-established that poly(ethylene oxide) (PEO) molecules in aqueous solutions are surrounded by a hydration layer similar to proteins.^{9,10} Moreover, it has recently been demonstrated by molecular dynamic (MD) simulations that stretching of oligomer PEO chains dissolved in water initiates the interchain aggregation, which ultimately leads to the phase separation of the PEO solution with the formation of highly oriented fibrillar nanostructures.¹¹ The aggregation was related to the change of PEO conformation making specific hydrogen-bond-induced solvation of PEO in water unfavourable which destroys the hydration layer. In this respect some previous observations of PEO fibrillation from aqueous solution under strong flows,¹²⁻¹⁴ assumed to be driven by PEO and water phase separation, could be explained by these recent MD simulations¹¹. Theoretical results indicate that solidification of PEO in water solution can be triggered by stretching in analogy with silk protein dopes. Using rheological properties of PEO and structural techniques based on birefringence and X-ray scattering, it is demonstrated herein that a simple synthetic polymer, PEO, with a

conformation-dependent hydration layer,⁹⁻¹¹ can be solidified and crystallised upon flow and processing conditions required for the flow-induced nucleation and crystallisation of the polymer are quantified and related to the molecular relaxation times.”

Term “shear” is used in our manuscript more than 70 times so clearly the flow type is known. In order to reduce number of words in the manuscript we tried to avoid term “shear flow” by using “shear” which is usually has the same meaning. Both types of flows (shear and extensional) are commonly used for studying flow-induced nucleation and crystallisation of polymers. We used techniques based on shear flow, which provides a better control of the flow parameters, and our results show that this type of flow worked perfectly well for chain stretching in our case.

Comment: 2. Furthermore, If they are trying to make a comparison with a silk solution, then why use a bimodal blend of 20K and 2 MDa PEO chains. Is there any evidence at all that silk solutions have bimodality? If not, then why not focus on a single molecular weight of high MW PEO in a suitably viscous solvent of say oligomeric Ethylene Glycol or Glycerol (i.e. a “PEO Boger fluid” as used by Pasquali, Macosko et al. over 20 years ago, AIChE J, 1998)? Wouldn't this be much closer to a silk protein solution? If not, then please explain why not?

Reply: In principle, silk dope is, at least, bimodal; it contains fibroin and sericin protein but we assume the referee referenced to a different bimodality. This comment misses the point of our study. The PEO must be processed in water. The observed phenomenon of flow-induced crystallisation is due to PEO-water interactions. We cannot change the solvent. This is the main point. Animals do not use Ethylene Glycol or Glycerol for processing proteins. Viscosity is not the only parameter responsible for crystallisation. In order to satisfy processing conditions of polymer to crystallise at ambient temperatures like animals do we have to deal with certain concentrations of PEO where PEO is in a metastable state. The required polymer concentrations are relatively high (50-60% w/w) and using only high molecular weight polymers at these concentrations will significantly increase viscosity of the fluid incomparable with animals having only 20% polymer (protein). Thus we dilute (mix) high molecular weight polymer in a low molecular weight matrix which enables the concentrated polymer solutions containing high molecular weight chains to be kept at a relatively low viscosity during processing. This point has been described in our original manuscript (lines 231-235): “In analogy with a previous research on thermoplastics^{24,25} a bimodal PEO blend was used. Blending a small fraction of long polymer chains (7.5% w/w, $M_w = 2$ MDa) with a polymer matrix of short chains (92.5% w/w, $M_w = 20$ kDa) allows the polymer crystal nucleation to be triggered at a relatively small flow rate, that is above inverse Rouse time, $\dot{\gamma}(t) > \dot{\gamma}_R = \tau_R^{-1}$, thereby stretching the long chains⁷ while maintaining a reasonably low viscosity.” In addition, the low molecular weight PEO in this system, crystallising on the flow-induced nuclei, works as an amplifier allowing the oriented morphology to be detected by structural techniques.

Comment: 3. The authors also make (rather unclear conjectural) arguments on p. 8 (lines 184-190) arguing for why polydispersity in their sample can lead to crystallization to occur at shear rates less than the estimated critical value. Why not just use monodisperse samples of high MW PEO. These are readily available as GPC standards and would allow a far clearer and more complete test of the theory that a critical shear rate (or Weissenberg number) does indeed have to be exceeded...

Reply: This is not a conjectural argument. We do not need to prove this point. It has been initially concluded in an extensive review on flow-induced crystallisation of polymers that a critical shear rate (close to reciprocal Rouse time of the longest chains in the polymer ensemble) has to be

exceeded (van Meerveld, Peters and Hutter, *Rheologica Acta*, 2004) and also clearly demonstrated in our work on flow-induced crystallisation of relatively monodisperse polyolefins (Mykhaylyk et al., *Macromolecules* 2008) and polydisperse polyolefins (*Macromolecules* 2010), each work cited more than 100 times, and the following work on trimodal blends (Okura et al, *Journal of Polymer Science B: Polymer Physics*, 2011). Our argument is based on these previously published works which are well recognised by community working on flow-induced crystallisation of polymers.

It would be hard, if possible at all, to make a monodisperse high molecular weight (such as 2MDa) PEO, even a standard will have a certain level of polydispersity. Flow works selectively for polymer chains and the longest molecules will be stretched first and, therefore, be in a favourable position to nucleate and as a result shift the experimentally observed threshold boundary towards smaller shear rates in a comparison with the expected one for the averaged ensemble of molecules. Considering the number of factors affecting the molecule Rouse time such as solvent concentration, polymer concentration, temperature, polymer polydispersity and the fact that Weissenberg number = 1 is only an approximate criterion it would be problematic to obtain a better correlation between the calculated Rouse time of 2MDa PEO (Supplementary Table 1) and the measured threshold shear rate (Figure 3A).

Comment: 4. In making their polymer physics arguments they make reference to the Milner/Likhtman/McLeish theory for chain-stretching in entangled solutions. This is clear and well-argued, but I wonder why they do not then compare their rheology data to the explicit prediction of the MLM theory? Why instead compare to an ad hoc empirical model (the Cross model) from the 1950s? This seems far from acceptable for a paper in *NatComm*?

Reply: The Milner/Likhtman/McLeish theory has been developed for relatively monodisperse polymers and not for polydisperse materials such as bimodal blends. The MLM theory is not applicable in our case to predict viscosity for a bimodal blend. If there is a concern, then a theory related to bimodal blends, such as presented by Rubinstein and Colby (*JCP*, 1988, p5291), should be used. However, in this case we would have to use an effective concentration and an effective molecular weight for the long-chain component because of the tube dilation of the long chains in the short chain matrix. Struglinski-Graessley parameter for the studied PEO system is well above the critical value of 0.064 (Park & Larson, *Macromolecules*, 2004, p597). These aspects for the bimodal blends have been discussed in our previous work (Okura et al, *PRL*, 2013). Considering all the above we have chosen the most reliable method – measure the viscosity experimentally at the same temperature as the polymer was processed during flow-induced nucleation (Figure 3B). The Cross model is simply used as an analytic expression to calculate the viscosity of samples in between the measured points. We could have equally used a simple linear equation between each pair of points.

Comment: 5. Furthermore, the Cross model they use is generally used for steady shear flow data not for *linear* viscoelastic measurements of $|\eta|$. Why do they do this? The flow they are considering is a strong steady shear flow, and they MUST make and report steady shear flow measurements of the rheological material response for both of their concentrated polymer systems. This should be readily achievable, and would obviate the need to invoke/assume (without any supporting evidence at all) the Cox-Merz rule (line 348). Also, why use “t” as one of the model parameters, since this is shearing time in their paper??

Reply: An application of a steady shear rheology is impossible in this case because the material will crystallise [as demonstrated by our rheo-SIPLI (Figure 3A and Supplementary Figure 1) and rheo-SAXS (Figure 4) for a steady shear] and the viscosity measurements of the polymer will not be reliable. We must avoid crystallisation in our measurements. That is why we have to use oscillatory

measurements and invoke Cox-Merz rule. The Cross model is used to fit the experimental viscosity data in order to obtain analytical expression for calculating viscosity used for the specific work calculations. We would like to thank the referee for spotting double usage of letter “t” for parameters. This mistake has been rectified in the revised manuscript, *t* was replaced by *k*.

Comment: 6. Although the arguments regarding shearing strength and time make both dimensional and physical strength –and are a plausible way of computing a specific shearing energy for crystallization – there is no evidence provided that the scalings proposed make physical sense. For example, it seems to immediately suggest to me that for each data point in Figure 3 one should do a test on the same fluid at twice the imposed shear rate, for half the imposed time. The results *must* (from their arguments) superpose. Do they?

Reply: We use a well-established parameter for characterising flow-induced nucleation - the specific work (or shearing strength as called by the reviewer). The parameter was originally introduced and physics behind this parameter was extensively discussed by Janeschitz-Kriegl’s group since 90s (see, for example, *Rheologica Acta* 2003, p355) and further confirmed and demonstrated by our group (Mykhaylyk et al, *Macromolecules* 2008 and *Macromolecules* 2010) that the specific work can be used as a parameter characterising flow-induced nucleation of a particular polymeric system as it is independent of shear rate and this approach has been used by other groups around the world (see, for example, Yang et al, *Macromolecular Rapid Communications*, 2017, 1700407). It was not a purpose of our current work to reintroduce the parameter and re-establish the methodology.

What the referee suggested in the comment is actually presented on Figure 3A. Each point on this plot corresponds to a boundary flow condition for the flow-induced nucleation measured for the same fluid but for a different time and for a different angular speed (the shear rate is controlled by the angular speed). Increase of shearing time reduces the shear rate required for nucleating the polymer. And what Figure 3C shows is that the specific work calculated from these parameters (including the viscosity to count the shear thinning as the referee rightly mentioned in their comment 8) is virtually constant and independent of shear rate at which nucleation is occurring. So to answer the referee’s question: Yes, they do.

Comment: 7. What about if they change the molecular weight of the long chain PEO? The Rouse time should change with Mw^2 , whereas the reptation time changes as Mw^3 (See their line 267). Which of these would best describe their experimental observations? Neither has been checked it seems to me? So why should we believe the chain stretching argument?

Reply: There is no need to believe. There are convincing experimental results. The supplementary Table 1 shows that the reptation (disengagement) time is nearly two or three orders of magnitude larger than the Rouse time. The threshold shear rate measured for the flow-induced nucleation in our work is about $10s^{-1}$ which is very close the reciprocal Rouse time of 2MDa PEO and clearly not to $\sim 0.01 s^{-1}$ (or less) required for the reptation mechanism. We reference the referee to our previous works (Okura et al, *Journal of Polymer Science B: Polymer Physics*, 2011 and Mykhaylyk et al, *European Polymer Journal*, 2011) where we have validated the method used by performing measurements on different molecular weights as proposed by the referee.

Comment: 8. Similarly it would seem that the scaling of the variation in shear rate across the plate must be investigated directly rather than just mentioned without experimental justification:

Although the shear rate will indeed vary linearly across the plate (as per their equation on 324-326) the shear stress will NOT (because of shear thinning driven by chain alignment and shear-induced convective constraint release (as explicitly predicted by the MLM framework). Therefore the local value of the specific work W will not vary linearly with position across the disk. This scaling is not mentioned or checked by the authors...and begs a very simple test; at fixed gap separation in the parallel plate device what happens to the crystallization boundary as they increase ω ? Does it shift in linearly with rate or in a more complex fashion? This would be very straightforward to check using images like Fig 3A at different speeds
For a paper in a high profile journal like NatComm I would expect some/all of these scalings to be checked...

Reply: The method used in our work for detecting flow conditions required for the flow-induced nucleation has already been extensively reported about a decade ago (Mykhaylyk et al., *Macromolecules* 2008 and Mykhaylyk et al., *Macromolecules* 2010) and referenced in our study (refs 7 and 21 in the original manuscript), it was not our intention to validate this method once again in the presented work. However, for the sake of consistency we have actually performed the scaling experiment proposed by the referee and the results are presented by Figure 3 of the original manuscript. As the referee pointed out the shear rate indeed varies linearly across the plate. But the specific work (W) is a product of shear rate and stress and, therefore, does not vary linearly with position across the disk by definition [see section “**Calculation of critical specific work for flow-induced crystallisation of PEO**”, lines 337-350]. It is not clear from the referee’s comment why it should. Each local value of W is defined by shear rate, stress (shear rate \times viscosity) and time. We totally agree with the referee that the shear stress will not vary linearly across the plate and the shear thinning is counted by viscosity (Figure 3C). The results of the simple test proposed by the referee are presented on Figures 3B and 3D. Each point on these plots represents a result obtained from an experiment similar to the one shown on Figure 3A but for a different angular speed. The position of the boundary moves but the specific work value corresponding to the measured boundary shear rate and time remains virtually constant (Figure 3D). Therefore, the validation proposed by the referee have already been performed and presented in the manuscript.

Comment: 9. Finally, regarding the analogy with silk spinning, it has been argued by Vollrath and Knight that an active ion-exchange is involved within the spinneret itself to drive extension-induced crystallization. No mention of this is given, and it naturally raises the question what would happen in the present system if the ionic strength (and PEO confirmation) was varied (e.g. using salt solutions or brine rather than DI water?)

Answer: Indeed, ion-exchange is used by animals, in particular, spiders to control properties of span silk fibres. However, our previous work together with Vollrath’s group (Holland et al, *Advanced Materials*, V24, 2012, 105-109) and another work from Vollrath’s group [Boulet-Audet et al, *Acta Biomaterialia*, V10, 2014, 776-784] show that crystallisation of silk under flow and the formation of fibres can take place without ion-exchange. Ion-exchange is an extra mechanism providing animals an additional control of their product. We have added a comment with an appropriate reference (ref 8) about ion-exchange in the revised manuscript on page 2: “In addition, there is an indication that animals can speed up the nucleation step by a careful control of the pH value and ion concentration in the processing environment.⁸”.

[Redacted]

Reviewer #3 (Remarks to the Author):

Comment: The authors present an interesting observation on the processing of PEO from an aqueous solution where the hydration shell of water surrounding the PEO is disrupted by the stretching of the PEO chains during flow, a breakage of the hydrogen bonds between the water and the PEO, that, in turn, lead to dehydrated segments that can form the nuclei of subsequent crystals. This is schematically shown in Figure 1. Initially, I had thought that this was a simulation but it is only an artist's rendering. Given the inability to probe this behavior and the subsequent initial aggregation to ultimately form nuclei, it raises the question as to why simulations were not done? Given the amount of speculation that is necessary to draw the conclusions (granted from several observations that infer this behavior), I would like to have seen some simulation support for the arguments. While the threshold shear rate to induced crystallization supports theoretical arguments, these arguments are based only on the stretching of the polymer chains and are devoid of the fundamental and perhaps the most important part of this work, namely the dehydration of the PEO chains. It is that, after all, that brings in the analogy to spiders their ability to retain the silk in a dissolved state, and the air ability to spin the fibers on demand.

Reply: Indeed, Figure 1 is an artist's rendering and it is indicated at the very beginning of the figure caption "A schematic representation...". However, it is not just an artist's rendering, this figure illustrates main conclusions made by recent molecular dynamic simulations [Donets, S. & Sommer, J.-U. Molecular Dynamics Simulations of Strain-Induced Phase Transition of Poly(ethylene oxide) in Water. *The Journal of Physical Chemistry B* **122**, 392-397, doi:10.1021/acs.jpcc.7b10793 (2018)] demonstrating that the sheath of water surrounding PEO can be destabilised by stretching the PEO molecules which leads to partial dehydration and aggregation (solidification) of the polymer chains. We have referenced this work in our original manuscript (**ref 10**) and cited four times in the text. The referee is absolutely correct that the most important part of this work is the dehydration of the PEO chains and it is that what brings the studied system in analogy to spiders capable to retain the silk in a dissolved state and spin the fibres on demand. Based on the theoretical MD simulations already published in 2018 we have designed a PEO system with appropriate rheological properties (described in Methods) to perform experiments confirming that namely stretching of PEO can initiate nucleation and crystallisation (solidification) of PEO in water. Following the referee's comment, we have extended discussion of this theoretical work in the main text of the revised manuscript (changed to **ref 11**) to make it clear that it is based on the simulation study: (**Page 2**) "It is well-established that poly(ethylene oxide) (PEO) molecules in aqueous solutions are surrounded by a hydration layer similar to proteins.^{9,10} Moreover, it has recently been demonstrated by molecular dynamic (MD) simulations that stretching of oligomer PEO chains dissolved in water initiates the interchain aggregation, which ultimately leads to the phase separation of the PEO solution with the formation of highly oriented fibrillar nanostructures.¹¹ The aggregation was related to the change of PEO conformation making specific hydrogen-bond-induced solvation of PEO in water unfavourable which destroys the hydration layer. In this respect some previous observations of PEO fibrillation from aqueous solution under strong flows,¹²⁻¹⁴ assumed to be driven by PEO and water phase separation, could be explained by these recent MD simulations¹¹. Theoretical results indicate that solidification of PEO in water solution can be triggered by stretching in analogy with silk protein dopes. Using rheological properties of PEO and structural techniques based on birefringence and X-ray scattering, it is demonstrated herein that a simple synthetic polymer, PEO, with a conformation-dependent hydration layer,⁹⁻¹¹ can be solidified and crystallised upon flow and processing conditions required for the flow-induced nucleation and crystallisation of the polymer are quantified and related to the molecular relaxation times." and (**Page 3**) "This sheath could be destabilised by a stimulus such as flow (**Fig. 1B**), stretching the PEO and leading to partial dehydration and aggregation of the polymer chains as predicted by MD simulations¹¹."

Comment: While I like this work overall as demonstration crystallization behavior of a hydrogen bonding polymer under shear, I am far, far less convinced of, what I consider to be hype, that his will lead to a new paradigm in polymer processing. Perhaps for a few selected, water soluble polymers but not form the preponderance of polymers that are currently processed and used on a large scale. Would the authors expect similar behavior similar behavior for conjugated polymer that exhibit ordering by intermolecular p-bonding to form H- or J-aggregates where the presence of solvents precludes interactions until a certain concentration range is reached? Here we have favorable interactions with the solvent and aggregation can occur only when the chains are in the proper conformation to interact. It's really not the same behavior but I have been trying to think of a situation where non-hydrogen bonding polymer would fall into the realm of this work. I was trying to come up with other example where such behavior could be observed. Do the authors have any suggestions to support the statement that this behavior could be a universal phenomenon, rather than simply making that statement?

Reply: The main goal of our work to find a synthetic analogue of aquamelt. We have designed a system and an experiment in order to test MD simulation results indicating that PEO aggregates in water solution upon stretching and our work actually demonstrates that stretching of PEO in aqueous solutions can initiate nucleation and crystallisation of PEO in a fashion somewhat similar to the silk processing used by spiders and silk worms. This is the first example where a synthetic polymer system shows such an interesting property and clearly this area requires a further exploration in order to find other examples. Our attempts with cellulose and cellulose derivatives where unsuccessful possibly due to stiffness of the polymer chains. We currently work on design and synthesis of new PEO (statistical) copolymer systems. Following the referee's comment, a sentence at the end of the original manuscript (Page 9, line 222) "This finding can lead to a new paradigm in polymer processing with vastly lower energy consumption." has been removed and a sentence at the end of the Abstract "This mechanism requiring vastly low energy consumption leads to a new paradigm in polymer processing." has been replaced by "This mechanism requires vastly lower energy consumption and demonstrates a new route for polymer processing." In addition, we made a suggestion at the end of the manuscript main text (Page 10) about what kind of systems should be considered as a potential candidate for aquamelt-type processing: "Considering the fact that PEO aqueous solutions show a stable cold crystallisation (as a result of water molecule bounding to the polymer), this phenomenon could be used as a criterion for selecting systems suitable for such a polymer processing. A similar suggestion was made for screening biocompatible polymer systems where a stable cold crystallisation had been proposed to use it as an index for binding water making the systems biocompatible.²⁵"

Comment: So, while I like the work, I would like to see some simulations to support the arguments made and, unless the authors present other firm examples, the hype in the manuscript should be substantially reduced.

Reply: The simulations supporting the argument have already been performed and published in scientific literature (see our reply to the first part of the referee's comment). Our experimental data confirm the simulation results and demonstrate that stretching of PEO initiates nucleation and crystallisation (solidification) of PEO in water indicating dehydration of PEO chains which we believe, and also recognised by the referee, is an important finding. Thus, both theoretical findings published in the literature and our experiments have a very good correlation with each other. The referee's comment about "the hype" has been appropriately addressed in the revised manuscript (see our reply to the previous comment).

Reviewers' comments:

Reviewer #1 (Remarks to the Author):

It seems the central question is whether the observed effect can really be viewed as analogous to spider silk solidification or not and whether it has something to do with changes in hydrogen bonding between PEO and water as shown schematically in Figure 1.

If the answer is no, then this work is more suitable for a more traditional polymer journals as the facts that a) PEO crystallizes in the presence of water, b) flow can stimulate crystallization in PEO melts and blends are well established and perhaps what is observed is just a change in a melting temperature induced by shear.

We already established that molecular forces for crystallization are different between PEO and silk, so the question remains whether the properties of the crystals are comparable or not. For instance, spider fibers do not dissolve in water (e.g. under rain) at room temperature, so do the produced PEO crystals remain stable at room condition in aqueous solution or do they dissolve and how quickly?

Based on the protocol used (Figure 2D) PEO solution firstly is heated up to produce dehydrated (phase separated) state, then quenched to 0 C for several minutes and then sheared to induce crystallization. To show that re-arrangement of water has something to do with observed effect one needs to establish what was the degree of hydration of PEO before shear is applied and what is it after, in the crystal state. It is not clear to me how the number of waters per PEO repeat unit are estimated. In the response the authors mention "that number was calculated from the masses of water and polymer mixed together" that assumes homogeneous mixing, right? But do we have such homogeneous mixing at 80 C or after the quench to 0C? Does water form ice and/or equilibrium between ice and liquid water at 0C in the presence of PEO? Are there any effects of the time the sample spends at high and low temperature before the shear pulse on the outcomes? If PEO is already sufficiently dehydrated before the shear then the schematics of Figure 1 has nothing to do with the observed effect.

Reviewer #2 (Remarks to the Author):

Revised version

The authors have done a nice job of addressing most of the comments – as well as explaining several points about the scaling analysis for calculating the specific work, as well as showing that they have already done some of the experiments suggested in earlier work. The revised introduction more clearly puts earlier work on PEO crystallization in context and connects to other recent studies. They have also toned down the hype related to "new paradigms for polymer processing", recognizing that this outlined mechanism will only work with a few very specific polar aqueous polymer solutions. I think this work can be accepted for Nature Communications subject to them addressing the remaining few residual comments/suggestions:

1. I think they should more clearly make the connection explicit that one must use dynamic viscosity measurements in conjunction with Cox-Merz rule – because if you try to actually measure the steady flow curve of stress vs rate you induce crystallization.
2. Line 378 is misleading I think...esp the use of "therefore".it is not because that the flow is laminar that the Cox-Merz relationship holds. One experiment is a steady shear flow, the other is a time-varying oscillatory flow; but both are laminar. It is a special result that works typically only (or best) for polydisperse systems. See comment 5 also.

3. The construct "modulus of complex viscosity" (e.g. text on p7/8, Fig3 and caption) is going to be confusing to people; even though it might be formally correct to refer to the modulus of a complex number, it is also fine to say "magnitude"; so why not write everywhere "magnitude of complex viscosity" or just use the $|\cdot|$ operator.

4. I still do not see the necessity of using the term "aquamelt" in the paper – this is most definitely not a melt, but a concentrated aqueous and bidisperse solution. I would recommend elimination of this jargon. The authors rebut that the term was "coined" – based on studies of silk solutions – to highlight materials that denaturize and have a solubilized metastable state. Nothing in this rebuttal suggests that "melt" is the right modifier to combine with aqua-xxx . Surely then referring to a "metastable aqueous polymer solution" is more accurate and scientific.

5. Regarding polydispersity: (original comment 3). Yes, I agree that even a PEO standard will have a certain level of polydispersity; but it is small and, perhaps, most importantly quantified and reported carefully (e.g. PDI < 1.01). No information at all is provided about the dispersity of the present samples except the vague statement on line 211 that they are "relatively polydisperse" . Relative to what??? We learn line134 that $M_w = 2\text{MDa}$ but we aren't told M_n or the polydispersity of these samples. Is it important? Or not? Surely a single clear definitive experiment with a monodisperse 2MDa sample would quantify how important the presence of some small fraction of even longer chains are in lowering the required shear threshold. At minimum, the molecular weight distribution of samples used MUST be characterized/measured and reported and the values of M_w/M_n provided.

The wording on line 214 also seems a little confusing: "...the shear rate low threshold limit..."? line 310 missing a "for". in "was accounted by correcting..."

Reviewer #3 (Remarks to the Author):

The authors have addressed the comments I raised in my review and, as i indicated in my earlier review, I feel this work will be an interesting contribution.

Reviewers' comments:

Reviewer #1 (Remarks to the Author):

Comment: It seems the central question is whether the observed effect can really be viewed as analogous to spider silk solidification or not and whether it has something to do with changes in hydrogen bonding between PEO and water as shown schematically in Figure 1.

If the answer is no, then this work is more suitable for a more traditional polymer journals as the facts that a) PEO crystallizes in the presence of water, b) flow can stimulate crystallization in PEO melts and blends are well established and perhaps what is observed is just a change in a melting temperature induced by shear.

Reply: PEO and silk protein water solutions are analogous - they both crystallise due to changes in hydrogen bonding with water molecules triggered by flow. So we cannot prove the difference. While crystallisation from solutions is widely observed by cooling a solution below its melting point and flow induced crystallisation is observed in polymer melts (which could be applicable to silk proteins as well), they have not been combined to trigger crystallization from a metastable solution. This differentiates our work from that of flow-induced crystallisation of thermoplastics or crystallisation from solution.

Flow can stimulate crystallisation of a PEO melt, but the mechanism is different from crystallisation in aqueous solution. It is not just the change in melting point by the addition of water. If this was the case PEO would crystallise as soon as it is cooled below its melting point. The novel point here is the presence of a hydration shell (a well-established fact appropriately referenced in our manuscript, refs 9, 10 and references therein) which keeps the polymer in a metastable state until it is perturbed by flow as demonstrated by the MD simulations (ref. 11). In this respect it is exactly the same as silk protein which is also surrounded by a hydration shell.

Comment: We already established that molecular forces for crystallization are different between PEO and silk, so the question remains whether the properties of the crystals are comparable or not. For instance, spider fibers do not dissolve in water (e.g. under rain) at room temperature, so do the produced PEO crystals remain stable at room condition in aqueous solution or do they dissolve and how quickly?

Reply: Flow-induced crystallisation of PEO aqueous solutions includes a few stages. At the initial stage of crystallisation (flow-induced nucleation) the molecular forces involved are mainly the same as in silk dope – changes in hydrogen bonding between water molecules and polymer molecules (PEO or silk protein) caused by flow. However, the locking mechanism solidifying the molecules is different, proteins form hydrogen bonds unperturbed by water while PEO molecules interact via Van der Waals force. As a result post-processing behaviour of PEO and silk is different: PEO dissolves in water at ambient conditions due to weak Van der Waals forces and silk does not due to strong hydrogen bonds. In our particular case PEO crystals remain solid after processing. This is demonstrated by shear-induced polarised light images (**Fig. 3A** and **Supplementary Fig. 2**) and X-ray

scattering patterns (**Fig. 4**) collected after processing. However, they dissolve upon adding water to the system as PEO crystals would normally do. We currently work on a design of statistical PEO copolymers in order to incorporate units with a capability to form hydrogen bonds after processing PEO aqueous solutions by shear. Silk is a superior material with a wide range of physical properties such as high toughness, high tensile strength, anti-bacterial properties etc. Perhaps, it would be rather impossible to reproduce all of these properties in a homogeneous synthetic polymer material. The important property which we replicate in a synthetic aqueous system is the change from a liquid to a solid triggered by flow. A property which no other synthetic material has achieved before.

Comment: Based on the protocol used (Figure 2D) PEO solution firstly is heated up to produce dehydrated (phase separated) state, then quenched to 0 C for several minutes and then sheared to induce crystallization. To show that re-arrangement of water has something to do with observed effect one needs to establish what was the degree of hydration of PEO before shear is applied and what is it after, in the crystal state. It is not clear to me how the number of waters per PEO repeat unit are estimated. In the response the authors mention "that number was calculated from the masses of water and polymer mixed together" that assumes homogeneous mixing, right? But do we have such homogeneous mixing at 80 C or after the quench to 0C? Does water form ice and/or equilibrium between ice and liquid water at 0C in the presence of PEO? Are there any effects of the time the sample spends at high and low temperature before the shear pulse on the outcomes? If PEO is already sufficiently dehydrated before the shear then the schematics of Figure 1 has nothing to do with the observed effect.

Reply: The Reviewer begins their comment with a statement about dehydration. This statement has nothing to do with our work. PEO is first heated to promote mixing of PEO and water to form a homogeneous solution. **We do not state at any point in the text that the solution is heated to dehydrate the PEO.** Moreover, in the previously revised manuscript an appropriately referenced text was added in the introduction (see page 3 of the previous manuscript) clearly explaining that a particular temperature interval had been chosen for our experiments in order to avoid the PEO dehydration: "PEO dehydration can also be stimulated by thermal treatments. An increase of temperature decreases the solvent quality through the reduction of hydrogen bonds; thereby PEO aqueous solutions undergo phase separation at a lower critical solution temperature of about 100 °C.^{22,23} Another example is cold crystallisation of PEO^{24,25} which is observed on heating preliminary cooled PEO-water mixtures where PEO is present in a glassy state and takes place at temperatures below solidus line of the PEO-water eutectic phase diagram²⁴ (about -21 °C). For these reasons the temperatures used for the shear experiments herein are above liquidus line of the PEO-water phase diagram and below 80 °C (**Methods**), and cannot stimulate the PEO and water phase separation without an external impact of flow." Thus, the temperature interval used in our work is inside of the temperature interval where the dehydration of PEO cannot occur because of temperature. We would also like to note that our experimental setup was designed in a way which almost completely prevented water evaporation. We made clear comments about this in the previous manuscript on page 12: "A sample was loaded by applying a small amount of PEO/water mixture from a syringe to the rheometer and then melted by heating to 80 °C in the presence of a saturated water atmosphere to prevent evaporation." and once more on page 14: "It is then cooled to the desired temperature (25 °C or 0 °C for 60% w/w or 50% w/w PEO solutions, respectively) and allowed to reach thermal equilibrium over 15 minutes in a saturated water atmosphere (the instrument is equipped with a solvent trap) to prevent evaporation." Since our experiments mainly use rheometers (and related

shearing devices), we performed monitoring of rheological properties (sensitive to phase separation, crystallisation and temperature equilibration) of the PEO solutions while selecting shear flow conditions and instrument setup for the performed experiments. In order to demonstrate the efficiency of our approach to stop water evaporation we have included data on magnitudes of complex viscosity measured with and without solvent trap in the revised manuscript (**Supplementary Fig. 8**) and appropriately referenced this in the revised text. The data convincingly show that there is a very pronounced effect of water evaporation from PEO solution if no extra measures, such as control of humidity by a solvent trap, are undertaken for our measurements. We have also added in the revised manuscript a sentence explaining why the heating step to 80 °C was used (see section “**Preparation and rheological properties of an aqueous solution of the linear PEO bimodal blend**” on page 12): “The heating step was used for homogenizing the PEO/water mixture before the shear pulse.”

In order to further demonstrate that there is no PEO dehydration caused by temperature treatments used in our experiments, we have performed FTIR spectroscopy of PEO aqueous solutions at different concentrations where a rheometer equipped with the solvent trap was used to provide the controlled environment preventing water loss by evaporation. An additional figure representing these results have been added in the revised manuscript (**Supplementary Fig. 1**). The C-O-C stretching band shifts from about 1097 cm^{-1} for 100% PEO to about 1083 cm^{-1} for PEO aqueous solutions upon adding of water (**Supplementary Figs. 1A and 1B**) which is attributed to the formation of hydrogen bonds between oxygen atoms in the ether backbone of PEO and water molecules. Accordingly, an extra text, appropriately referenced, has been added in the revised manuscript on page 2: “and Fourier-transform infrared (FTIR) spectroscopy where there is an effect of the bound water on the frequency of PEO ether (C-O-C) stretching band peak at 1080-1100 cm^{-1} (**Supplementary Fig. 1 and Methods**). In molten (anhydrous) PEO the ether stretching band is observed at 1097 cm^{-1} whereas in well-solvated, dilute PEO (such as PEO in 50% w/w aqueous solution) the ether stretching band is observed at 1084 cm^{-1} (**Supplementary Fig. 1A**). As water is added to the PEO the peak position falls rapidly reaching the limiting solvated value at concentrations around 60% w/w (**Supplementary Fig. 1B**). The decrease in the C-O-C stretching frequency upon adding of water is commonly attributed to the formation of hydrogen bonds between oxygen atoms in the ether backbone of PEO and water molecules.¹⁹” as well as in the “Methods” section for a description of the performed experiments. The FTIR results (the ether stretching band peak position, **Supplementary Figs. 1B and 1D**, and its intensity, **Supplementary Figs. 1C and 1E**) show that the PEO does not change its hydrated state at 80 °C over the duration of heat-treatment used in our experiments. In this respect a comment on page 12 of the revised manuscript has been added: “FTIR spectroscopy monitoring of PEO aqueous solutions at the elevated temperature indicated that there was no effect of this treatment on the sample composition (**Supplementary Figs. 1D and 1E**).” Thus, neither water evaporation nor dehydration caused by heat-treatments at elevated temperatures (up to 80 °C), which could affect the composition of PEO aqueous solution, take place in the experiments performed in our study.

Finally, we have performed SIPLI measurements of flow-induced nucleation of 50% w/w PEO aqueous solution without a heating step to 80 °C. The sample was loaded in the SIPLI rheometer at room temperature (21 °C) and then cooled straight down to 0 °C before applying a shear pulse to induce the nucleation (**Fig. R1, see below**) using a shear flow similar to the experiment presented in the manuscript (**Fig. 3A**). The SIPLI results show (**Fig. R1, see below**) that crystallisation of PEO can

be induced with a similar boundary flow conditions without the heating step to 80 °C associated by the Reviewer with PEO dehydration. We would also like to note that the presented polarised light images show a few bright streak bands along the flow directions in the middle of the sample which is a result of some local inhomogeneity of the PEO/water mixture caused by a shear during loading the sample (which is also a common problem for silk dope transfers). In order to avoid this, the heating step to 80 °C homogenising a PEO/water sample was introduced in our experiments (heating cannot be used for a silk dope because of protein denature) before applying a shear pulse. Thus, once more, we have added in the revised manuscript a sentence clarifying why the heating step to 80 °C was used (see section “**Preparation and rheological properties of an aqueous solution of the linear PEO bimodal blend**” on page 12): “The heating step was used for homogenizing the PEO/water mixture before the shear pulse.”

Fig. R1: Representative polarised light images (circular images) of a sheared 50% w/w PEO aqueous solution recorded during a SIPLI experiment ($R = 12.5$ mm and SIPLI image diameter = 25 mm, $d = 0.5$ mm, $\omega = 4$ rad/s, and $t_p = 23$ s), with a time-lapse (rectangular image) composed of 45° slices through images over the course of the measurement. The sample area enclosed between dot-dashed and double-dot-dashed circles in the last image of the experiment corresponds to PEO crystallised by flow-induced nucleation.

We take special care to prepare homogenous PEO/water mixtures. We heat the mixtures and stir several times over the course of about 24 hours before the experiment to ensure homogeneous mixing which is explicitly described in our original manuscript in section “**Preparation and rheological properties of an aqueous solution of the linear PEO bimodal blend**”: “To form aqueous PEO solutions (50% w/w and 60% w/w) sheets of the blended PEO (100% w/w) where cut into strips

and placed inside a polypropylene disposable syringe (10ml) followed by adding the required mass of water. The syringe tube was then sealed by attaching a plug to the needle hole before being heated to 70 °C to aid mixing. The PEO strips had visibly dissolved after a few hours. To homogenize the samples a second syringe was attached to the first and the mixture pumped back and forth between the tubes several times. The syringe tubes were then allowed to stand overnight to equilibrate the distribution of water, before being centrifuged at 4,000 g for 10 minutes to remove any air bubbles. These aqueous solutions of bimodal PEO blends were then used in subsequent experiments." X-ray scattering of the mixture shows no crystalline peaks, and DSC measurements give a single crystallisation/melting peak. Therefore, the mixture is homogenous and the degree of hydration can be calculated from the masses of water and PEO mixed together to be ~1.6 water per monomer unit (according to 60% w/w PEO aqueous solution composition, where the effect of water on PEO crystallisation becomes clearly detectable by DSC results, see **Fig. 2C**). Following shear, x-ray scattering of the processed material shows patterns consistent with PEO crystals produced in the absence of water indicating that water is not included in the crystal structure; therefore, the degree of PEO hydration is ~0 (we are not aware of any work demonstrating opposite). So during the flow-induced crystallisation the PEO hydration state has changed from 1.6 to 0.

Water does not freeze in the presence of 50% PEO within the temperature interval used for the shear experiments. This is indicated by our DSC measurements (**Fig. 2B**) as well as by other works studying PEO-water system and its phase diagram (see refs 25 and 26 in our manuscript). If water crystals did form, they would be clearly visible in the x-ray scattering patterns (or in DSC), which they are not.

There is no effect of the time the sample spends at high temperature unless the temperature is above LCST (about 100 °C) or water evaporation occurs. Our measurements were performed in a controlled humidity environment to stop water evaporation. Without this precaution the sample composition and, consequently, shear results would change. In respect to the low temperatures, there is no time effect if the sample is kept at temperatures above equilibrium melting point of the PEO/water mixture in a controlled humidity environment. Also viscosity monitoring was performed at low temperature to make sure that the sample state does not change in order to set the time delay for temperature equilibration before the shear pulse (see **Supplementary Fig. 8** added to the revised manuscript). However, if the sample is kept below the melting point too long (several hours) thermal nucleation occurs which will affect the flow-induced nucleation as it was indicated in the original manuscript (see page 8): "At such high undercooling, however, the PEO solutions are much more susceptible to thermal nucleation and have to be used in a shorter period (within hours)."

As demonstrated by our reply, design of our experiment and chosen temperature interval excludes sufficient dehydration of PEO and, thus, results reported in our manuscript strongly support MD simulation results summarised by Figure 1.

Reviewer #2 (Remarks to the Author):

Revised version

Comment: The authors have done a nice job of addressing most of the comments – as well as explaining several points about the scaling analysis for calculating the specific work, as well as showing that they have already done some of the experiments suggested in earlier work. The revised introduction more clearly puts earlier work on PEO crystallization in context and connects to other recent studies. They have also toned down the hype related to “new paradigms for polymer processing”, recognizing that this outlined mechanism will only work with a few very specific polar aqueous polymer solutions. I think this work can be accepted for Nature Communications subject to them addressing the remaining few residual comments/suggestions:

Reply: We would like to thank you the Reviewer for this very positive feedback.

Comment: 1. I think they should more clearly make the connection explicit that one must use dynamic viscosity measurements in conjunction with Cox-Merz rule – because if you try to actually measure the steady flow curve of stress vs rate you induce crystallization.

Reply: Thank you for this suggestion. We have clarified this point in the revised manuscript on page 15: “Since steady-state viscosity measurements of aqueous PEO solutions at flow-induced nucleation conditions is complicated by an initiation of the polymer crystallisation, dynamic viscosity measurements have been performed instead (Fig. 3C and Supplementary Fig. 5) and the Cross model fitted to the experimental data (Fig. 3C and Supplementary Fig. 5, right column).”

Comment: 2. Line 378 is misleading I think...esp the use of “therefore”.it is not because that the flow is laminar that the Cox-Merz relationship holds. One experiment is a steady shear flow, the other is a time-varying oscillatory flow; but both are laminar. It is a special result that works typically only (or best) for polydisperse systems. See comment 5 also.

Reply: “therefore” has been removed from the text and the corresponding phrase was changed in the revised version from “...the material experienced a laminar flow during the shear pulse and, therefore,...” to “...there were no transient effects due to flow and...”.

Comment: 3. The construct “modulus of complex viscosity” (e.g. text on p7/8, Fig3 and caption) is going to be confusing to people; even though it might be formally correct to refer to the modulus of a complex number, it is also fine to say “magnitude”; so why not write everywhere “magnitude of complex viscosity” or just use the $| |$ operator.

Reply: Done. “modulus” has been replaced by “magnitude” through the entire manuscript text.

Comment: 4. I still do not see the necessity of using the term “aquamelt” in the paper – this is most definitely not a melt, but a concentrated aqueous and bidisperse solution. I would recommend elimination of this jargon. The authors rebut that the term was “coined” – based on studies of silk solutions – to highlight materials that denaturize and have a solubilized metastable state. Nothing in this rebuttal suggests that “melt” is the right modifier to combine with aqua-xxx . Surely then referring to a “metastable aqueous polymer solution” is more accurate and scientific.

Reply: As suggested by the Reviewer term “aquamelt” has been appropriately replaced by “metastable aqueous polymer solution” in the revised manuscript.

Comment: 5. Regarding polydispersity: (original comment 3). Yes, I agree that even a PEO standard will have a certain level of polydispersity; but it is small and, perhaps, most importantly quantified and reported carefully (e.g. PDI < 1.01). No information at all is provided about the dispersity of the present samples except the vague statement on line 211 that they are “relatively polydisperse” . Relative to what??? We learn line134 that $M_w = 2\text{MDa}$ but we aren’t told M_n or the polydispersity of these samples. Is it important? Or not? Surely a single clear definitive experiment with a monodisperse 2MDa sample would quantify how important the presence of some small fraction of even longer chains are in lowering the required shear threshold. At minimum, the molecular weight distribution of samples used MUST be characterized/measured and reported and the values of M_w/M_n provided.

Reply: Following the Reviewer comments the molecular weight distribution of PEO samples has been measured and the results (size exclusion chromatograms, number-average molecular weight M_n , weight-average molecular weight M_w , higher-weight-average molecular weight M_z and M_w/M_n) are included in the revised manuscript (see section “**Preparation and rheological properties of an aqueous solution of the linear PEO bimodal blend**” on page 11 and **Supplementary Fig. 4**). Rouse time and critical shear rate values previously calculated for nominal molecular weights of PEO have been recalculated accordingly for the measured values of M_w and M_z (**Supplementary Table 1**). The new values are close to the previous values and do not affect the discussion and conclusions made in the manuscript. Moreover, the results show that there is a good correlation between the estimated critical shear rate for stretching the higher-weight-average molecular weight of PEO and the shear rate threshold observed experimentally for the flow-induced nucleation of PEO. This fact has been appropriately commented in the revised manuscript on page 10: “In particular, $\dot{\gamma}_{RC}$ estimated for the PEO molecules with higher-weight-average molecular weight $M_z = 2851\text{ kDa}$, indicative of higher molecular weight polymers present in the polymer ensemble, is similar to the experimentally detected value of the lowest shear rate resulting in PEO flow-induced nucleation (**Fig. 3B**).”

Comment: The wording on line 214 also seems a little confusing: “...the shear rate low threshold limit...”?

Reply: Done. The “...the shear rate low threshold limit...” was replaced by “...the lowest shear rate...” and the sentence was appropriately adjusted.

Comment: line 310 missing a "for". in "was accounted by correcting..."

Reply: Done.

Reviewer #3 (Remarks to the Author):

Comment: The authors have addressed the comments I raised in my review and, as i indicated in my earlier review, I feel this work will be an interesting contribution.

Reply: We would like to thank you the Reviewer for supporting our work.

REVIEWERS' COMMENTS:

Reviewer #1 (Remarks to the Author):

In the revised manuscript the authors clarified several points. For instance, they made clear that PEO crystals will dissolve in water (in contrast to silk) and ensured that PEO solution is not phase separated (at least to the extent of having more than one peak in DSC) before shearing. Yet, other major issues remain unanswered. In my previous comments I summarized the main issue, which in my opinion is the following: "It seems the central question is whether the observed effect can really be viewed as analogous to spider silk solidification or not and whether it has something to do with changes in hydrogen bonding between PEO and water as shown schematically in Figure 1. " In the authors response they state that the analogy is that PEO can crystalize from solution under flow, as silk, which is the novelty of the manuscript. However, the examples of flow-induced crystallization of polymers from solutions have been studied and reported in literature – see e.g. S. Matsuzawa, et al., *Kolloid-Z.u.Z.Polymere* 250, 20–26 (1972) for crystallization of poly(vinyl alcohol) from its aqueous solution under flow or Z. Pelzbauer et al., *Journal of Macromolecular Science, Part B*, 4:4, 761-773, 1970 for PEO fibrillar crystals prepared from ethyl alcohol solutions by stirred crystallization. These examples show polymer crystallization from solution by shearing is not a new phenomenon and not limited to PEO and silk and may not necessarily require a hydration shell, as it can be observed for PEO in a different solvent. Furthermore, authors do not prove that there is no water in the obtained PEO crystal. They state that "Following shear, x-ray scattering of the processed material shows patterns consistent with PEO crystals produced in the absence of water indicating that water is not included in the crystal structure; therefore, the degree of PEO hydration is ~ 0 (we are not aware of any work demonstrating opposite)". Yet, it is known that presence of water (e.g. up to 12wt% can come from air humidity) is actually favorable for PEO crystallization, so similarity of peaks in crystal structure may not necessary confirm that hydration is zero. Thus, I do not think comparison with silk is justified and/or the concept of Figure 1 is proven.

Overall, the paper reports a nicely done experiment with interesting observations and could be considered for publication by the editorial office if comparison with silk is removed (or minimized) as misleading; the appropriate references on other examples of polymer crystallization from solutions included in the literature review and other explanations for the observed effect besides of the idea presented in Figure 1 are discussed (e.g. orientation/elongation-induced crystallization).

Reviewer #2 (Remarks to the Author):

The authors have done a good job in addressing all of my concerns. Although I suspect that very few synthetic/commercial polymer systems will be able to be processed this way, I do think this will make an interesting contribution.

One tiny edit: on p11 when they say "triggered at a relatively small flow rate" and then give a criterion, the criterion is (correctly given as a shear rate or inverse of the Rouse time. It would thus be more proper to write "triggered at a relatively small SHEAR RATE..."

REVIEWERS' COMMENTS:

Reviewer #1 (Remarks to the Author):

Comment: In the revised manuscript the authors clarified several points. For instance, they made clear that PEO crystals will dissolve in water (in contrast to silk) and ensured that PEO solution is not phase separated (at least to the extent of having more than one peak in DSC) before shearing. Yet, other major issues remain unanswered.

Reply: We are glad to see that resolution is being reached in the areas the reviewer has highlighted. Although we disagree with some of their points, we wish to thank the reviewer for their effort making the manuscript as strong as possible.

Comment: In my previous comments I summarized the main issue, which in my opinion is the following: "It seems the central question is whether the observed effect can really be viewed as analogous to spider silk solidification or not and whether it has something to do with changes in hydrogen bonding between PEO and water as shown schematically in Figure 1. " In the authors response they state that the analogy is that PEO can crystallize from solution under flow, as silk, which is the novelty of the manuscript. However, the examples of flow-induced crystallization of polymers from solutions have been studied and reported in literature – see e.g. S. Matsuzawa, et al., *Kolloid-Z.u.Z.Polymer* 250, 20–26 (1972) for crystallization of poly(vinyl alcohol) from its aqueous solution under flow or Z. Pelzbauer et al., *Journal of Macromolecular Science, Part B*, 4:4, 761-773, 1970 for PEO fibrillar crystals prepared from ethyl alcohol solutions by stirred crystallization. These examples show polymer crystallization from solution by shearing is not a new phenomenon and not limited to PEO and silk and may not necessarily require a hydration shell, as it can be observed for PEO in a different solvent.

Reply: We totally agree with the Reviewer that polymer crystallisation from solution by shearing (actually by stirring) is not a new phenomenon. We have clearly stated this in the introduction of the manuscript while discussing mechanism of PEO crystallisation from aqueous solutions (page 3): "This proposed mechanism is qualitatively different from the shish-kebab formation demonstrated for polyethylene deposited on a stirrer surface during shear of supercooled polyethylene-xylene solution at temperatures above 100 °C.^{21,22}" In this respect, work by Z. Pelzbauer et al. (cited by the Reviewer) also uses a stirrer to produce PEO crystals where stretched PEO molecules crystallise on (the free surface of) a stirrer which is explicitly stated at the beginning of the paper: "The crystallized polymer collected on the stirrer and, after re-recovery, was washed in fresh ethanol at the temperature of crystallization. Subsequently it was washed in ether at room temperature and dried under vacuum." Furthermore, the authors of this work appeal to a paper by A. J. Pennings, *J. Polymer Sci.*, C16, 1799 (1967) reporting a similar observation for polyethylene crystallisation on a stirrer from a xylene solution which is essentially the same work cited in our manuscript where we referenced an extensive review, summarising these phenomena, published later by the same author [Ref 21: Pennings, A. J. Bundle-Like Nucleation and Longitudinal Growth of Fibrillar Polymer Crystals from Flowing Solutions. *Journal of Polymer Science Part C-Polymer Symposium*, 55-86 (1977)]. The other work cited by the Reviewer [S. Matsuzawa, et al., *Kolloid-Z.u.Z.Polymer* 250, 20–26 (1972)] is also analogous in this respect. However, no crystal structure analysis was performed and no other structural characterisation techniques were used in this work so we prefer to refrain from further comments.

In order to emphasise our point made in the discussion we have revised the relevant sentence and included an additional reference to work by Z. Pelzbauer et al. [ref 23 in the revised manuscript]

suggested in the reviewer's comment: "This proposed mechanism is qualitatively different from the shish-kebab formation demonstrated for polymers deposited on a free surface during the stirring of solutions of supercooled polyethylene-xylene^{21,22} or PEO-ethanol²³ solutions."

Comment: Furthermore, authors do not prove that there is no water in the obtained PEO crystal. They state that "Following shear, x-ray scattering of the processed material shows patterns consistent with PEO crystals produced in the absence of water indicating that water is not included in the crystal structure; therefore, the degree of PEO hydration is ~0 (we are not aware of any work demonstrating opposite)". Yet, it is known that presence of water (e.g. up to 12wt% can come from air humidity) is actually favorable for PEO crystallization, so similarity of peaks in crystal structure may not necessary confirm that hydration is zero. Thus, I do not think comparison with silk is justified and/or the concept of Figure 1 is proven.

Reply: If presence of water is favourable for PEO crystallisation it does not necessarily mean that water molecules are accommodated in the PEO crystal structure. We would like to reiterate that our previous statement "we are not aware of any work demonstrating opposite" in reply to Reviewer 1 is related to the PEO crystal structure and not to crystallisation. The PEO crystal structure formed from water solutions in our experiment is consistent with an anhydrous 7/2 helix crystal structure, the same as that formed by PEO from the melt. If the crystal structure were to accommodate an extra 12wt% of water this would be clearly noticeable in x-ray scattering patterns as an increase in unit cell dimensions or even a change in the unit cell symmetry. Neither of these are observed. Moreover, works on crystal structure of PEO [Ref 20: Takahashi, Y. & Tadokoro, *Macromolecules* **6**, 672-675 (1973) (7/2 helix); H. Tadokoro et al., *Makromol. Chem.* 1964, 73, 109 or A. C. French et al., *Angew. Chem. Int. Ed.*, 48, 1248-1252 (2009) (3/10 helix); Takahashi, Y et al., *Journal of Polymer Science: Polymer Physics Edition*, 11, 2113-2122 (1973) (zig-zag similar to polyethylene)] do not report any water molecules present in the PEO crystal structure. We also are not aware of any published work on PEO demonstrating incorporation of water molecules in PEO 7/2 helix crystal structure and the Reviewer 1 does not provide any such evidence. Furthermore, the authors of the crystallographic work reporting 7/2 helical structure (Takahashi, Y. & Tadokoro, *Macromolecules*, 1973) indicated that this structure is formed due to flexibility of PEO chains and intermolecular PEO-PEO forces and this was explicitly referenced in our manuscript: "When the stretching is released the dehydrated PEO chains are likely to relax into their stable 7₂ helical crystal structure (space group P2₁/a) because of chain flexibility and the PEO-PEO intermolecular forces²⁰". It has to be noted that in one of the works referenced in our manuscript [Ref 10: Dahal, U. R. & Dormidontova, E. E. *The dynamics of solvation dictates the conformation of polyethylene oxide in aqueous, isobutyric acid and binary solutions. Physical Chemistry Chemical Physics* **19**, 9823-9832 (2017)] it was commented that "PEO can form a 3/10 helix structure in the crystalline state with a small inclusion of water". However, diffraction pattern of 3/10 helix crystal structure is significantly different from 7/2 helix crystal structure observed in our experiments (Figure 4, WAXS inset). Thus, observation of the 7/2 helix in our experiment after shear and crystallisation is a compelling argument that the hydration shell (actually, at least 40wt% and not just up to 12wt%) has been destroyed by shear and removed from PEO molecules and is therefore analogous behaviour to silk. Based on the fact that both silk protein and PEO have a hydration shell, and that in both cases this shell can be removed by a shear flow that triggers crystallisation, a comparison of PEO with silk is well justified. The concept of Figure 1 based on MD simulations [Ref 13: Donets, S. & Sommer, J.-U. *Molecular Dynamics Simulations of Strain-Induced Phase Transition of Poly(ethylene oxide) in Water. The Journal of Physical Chemistry B* **122**, 392-397, (2018)] which is also partially confirmed by experimental results [Ref 29: Liese, S. *et al. Hydration Effects Turn a Highly Stretched Polymer from an Entropic into an Energetic Spring. ACS*

Nano 11, 702-712, (2017)] is proven by the experimental results presented in our manuscript. Thus, our manuscript related to this part of the work remains unchanged.

Comment: Overall, the paper reports a nicely done experiment with interesting observations and could be considered for publication by the editorial office if comparison with silk is removed (or minimized) as misleading; the appropriate references on other examples of polymer crystallization from solutions included in the literature review and other explanations for the observed effect besides of the idea presented in Figure 1 are discussed (e.g. orientation/elongation-induced crystallization).

Reply: We are grateful to the Reviewer 1 for giving some credit to our work. As we have pointed out in our reply to the reviewer's previous comments the comparison with silk, based on the experimental observations in our work and previous experimental and theoretical results, is well justified and not misleading. This piece of research was inspired by the observations that silk required 2 orders of magnitude less work to induce crystallisation than a conventional polymer melt. The PEO-water system was selected for study because of its particular phase behaviour, and demonstrates the similarities to silks response to flow that were predicted by the molecular dynamic simulations referenced herein. Thus, this part of the manuscript remains mainly unchanged. However, to address the reviewer's comment about over emphasis of the comparison with silk we have changed the title of the paper and removed "silk" from the title: **"Flow-Induced Crystallisation of Polymers from Aqueous Solution"**.

In response to the reviewer's comment about polymer crystallisation from stirred solutions we have extended our discussion of the previous work in the manuscript and added a reference cited by the Reviewer 1 (**Ref 23 in the revised manuscript**).

The idea of orientation/elongation –induced crystallisation is discussed in both the Introduction section and in the Methods section using a number of references to the scientific literature broadly covering the subject: **Refs 3, 6, 7, 21, 22, 23, 28, 29, 32, 33, 35 and 38**. Flow-induced crystallisation has already been extensively discussed in our response to the comments of Reviewer 2 during first round of reviews, this subject has been appropriately addressed in the revised manuscript. However, following the reviewer's comment we have extended our discussion further by comparing the observed phenomena with flow-induced crystallisation of polymer melts including alignment/orientation (see Page 10, end of the first paragraph of "Discussion" section):

"There are many similarities between the flow-induced nucleation and crystallisation of polymers from aqueous solution and the flow-induced crystallisation of thermoplastics from the melt state. For polymer melts the process can be subdivided into three stages: stretching, nucleation and alignment of the nuclei formed.³³ The stretching introduces conformational order into the polymer chains, reducing the energy barrier for nucleation, and flow delivers one stretched segment to another until they collide and form an aggregate which is larger than the critical size of a stable nucleus. Once formed, the nuclei align along the flow direction and oriented crystals grow. The same three stages are also seen in this study of aqueous polymer solutions with one crucial difference: in this case the stretching process not only induces conformational order but also removes the solvent sheath and, therefore, reduces two barriers to polymer nucleation and subsequent crystallisation. To enable crystallisation both barriers must be affected by flow, it is not sufficient to just stretch the polymer in solution, the solvent sheath must also be removed allowing the stretched chains to aggregate."

Reviewer #2 (Remarks to the Author):

Comment: The authors have done a good job in addressing all of my concerns. Although I suspect that very few synthetic/commercial polymer systems will be able to be processed this way, I do think this will make an interesting contribution.

Reply: We are grateful to the Reviewer for their comments and high ranking of our work.

Comment: One tiny edit: on p11 when they say "triggered at a relatively small flow rate" and then give a criterion, the criterion is (correctly given as a shear rate or inverse of the Rouse time. It would thus be more proper to write "triggered at a relatively small SHEAR RATE..."

Reply: Done.